# Unifying the analysis of bottom-up proteomics data with CHIMERYS

**Martin Frejno** [1,4] ✉, **Michelle T. Berger** [1,4], **Johanna Tüshaus** [2,4], **Alexander Hogrebe** [1,4], **Florian Seefried**[1], **Michael Graber**[1], **Patroklos Samaras**[1], **Samia Ben Fredj**[1], **Vishal Sukumar**[1], **Layla Eljagh**[1], **Igor Bronshtein**[1], **Lizi Mamisashvili**[1], **Markus Schneider**[1], **Siegfried Gessulat**[1], **Tobias Schmidt** [1], **Bernhard Kuster** [2,3], **Daniel P. Zolg**[1] & **Mathias Wilhelm** [2,3] ✉

Proteomic workflows generate vastly complex peptide mixtures that are analyzed by liquid chromatography–tandem mass spectrometry, creating thousands of spectra, most of which are chimeric and contain fragment ions from more than one peptide. Because of differences in data acquisition strategies such as data-dependent, data-independent or parallel reaction monitoring, separate software packages employing different analysis concepts are used for peptide identification and quantification, even though the underlying information is principally the same. Here, we introduce CHIMERYS, a spectrum-centric search algorithm designed for the deconvolution of chimeric spectra that unifies proteomic data analysis. Using accurate predictions of peptide retention time, fragment ion intensities and applying regularized linear regression, it explains as much fragment ion intensity as possible with as few peptides as possible. Together with rigorous false discovery rate control, CHIMERYS accurately identifies and quantifies multiple peptides per tandem mass spectrum in data-dependent, data-independent or parallel reaction monitoring experiments.

Mass spectrometry (MS)-based bottom-up proteomics is the mainstay technology for high-throughput protein identification and quantification today[1–3]. The former is achieved by matching theoretical, predicted or library fragment ion mass spectra (MS2) to experimental MS2 spectra, which contain sequence and amino acid modification information on peptide precursor ions, measured in precursor mass spectra (MS1). Today, MS2 spectra are typically acquired in data-dependent (DDA), data-independent (DIA) or parallel reaction monitoring (PRM) mode. Peptide quantification either uses the precursor intensity from MS1 (DDA) or fragment ion intensities from MS2 (DIA and PRM) spectra. A central challenge for data analysis is the fact that most MS2 spectra are chimeric (they contain more than one peptide)[4–6]. This is because liquid chromatography–tandem MS (LC–MS/MS) systems cannot fully separate the vast number of peptides resulting from whole proteome enzymatic digestion, in particular when short gradients or no liquid chromatography at all is employed, as exemplified by direct infusion-shotgun proteome analyses (DI-SPA)[7].

DIA MS2 spectra are usually more complex than DDA MS2 spectra because they are typically acquired with wider isolation windows to maintain low MS cycle times (important for quantification) and hence contain fragment ions from many different precursors[8]. Although DDA and PRM MS2 spectra are normally acquired to minimize co-isolation, they are also chimeric, albeit to a lesser extent[6]. Because of the way that data acquisition approaches have evolved, the corresponding data types are analyzed differently[9], making it difficult to compare them in an unbiased fashion[10].

[1]MSAID GmbH, Garching b. München, Germany. [2]School of Life Sciences, Technical University of Munich, Freising, Germany. [3]Munich Data Science Institute (MDSI), Technical University of Munich, Garching b. München, Germany. [4]These authors contributed equally: Martin Frejno, Michelle T. Berger, Johanna Tüshaus, Alexander Hogrebe. ✉e-mail: martin.frejno@msaid.de; mathias.wilhelm@tum.de

DDA data are analyzed in a spectrum-centric fashion[9]. Database search algorithms for DDA data attempt to maximize identifications from chimeric spectra by submitting them for each precursor detected in the isolation window. Frequently, fragment ions explained by a given peptide are removed from the spectrum before it is searched again in a subtractive approach[6,11]. While often able to identify multiple peptides, this approach under-utilizes spectral information when fragments are shared between peptides, resulting in reduced sensitivity. When fragment ions are not removed before an additional search (multiplicative approach), the same information may be used too often, resulting in reduced specificity. In the end, the central output of DDA search engines is one or multiple peptide-spectrum matches (PSMs) per experimental MS2 spectrum.

In contrast, DIA and PRM data analysis usually follows a peptide-centric approach that asks the question whether peptides from a pre-defined list are detectable in the experimental data[9,12]. This approach requires spectral libraries, which can be generated from previous experimental data or predicted via machine or deep-learning models. Subsequently, the queried peptides are detected and quantified in MS1 and/or MS2 spectra by extracting co-eluting (fragment) ion chromatograms (XICs) based on the spectral library. Recently, library-free approaches such as DIA-Umpire[13], PECAN[14], directDIA[15] (implemented in Spectronaut), MSFragger-DIA[16] and diaTracer[17] gained popularity due to their simplicity. In brief, these tools do not require the generation of a spectral library and instead identify peptides in DIA data given a set of query peptides by directly scoring experimental MS2 or 'pseudo-MS/MS' spectra against theoretical spectra.

Because of the molecular complexity of proteomic samples and the large quantities of MS2 spectra of varying quality that are generated by LC–MS/MS, accurate false discovery rate (FDR) control is important, particularly in large-scale projects. While FDR control for DDA data is rather mature[18–21], it is still a substantial challenge for DIA data. Constructing realistic decoy MS2 spectra and retention times is far from obvious, an issue increasingly realized and addressed by machine-learning models for peptide property prediction[22–24].

In this work, we introduce a spectrum-centric and data acquisition method-agnostic algorithm for the analysis of MS2 spectra, implemented in CHIMERYS. It deconvolutes any MS2 spectrum, regardless of whether it was acquired by DDA, DIA or PRM, thus unifying the analysis of bottom-up proteomics data. We build upon a concept introduced for the deconvolution of DIA spectra using spectral libraries[8] and leverage deep-learning-based predictions of fragment ion intensities from INFERYS[25] in conjunction with linear algebra for the deconvolution of MS2 spectra. The resulting signal contributions of each peptide identified in each MS2 spectrum can be combined into a quantitative readout. Applying the approach substantially enhances identification rates of PSMs, peptides, and proteins across all sample types in DDA, enables the hands-off processing of PRM data and matches the performance of alternative DIA software while maintaining accurate FDR control throughout.

## Results

### Deconvolution of chimeric DDA spectra

The core assumption behind CHIMERYS is that chimeric MS2 spectra are linear combinations of pure spectra from co-isolated precursors. The algorithm is entirely spectrum-centric and employs non-negative L1-regularized regression via the LASSO[26] to explain as much experimental intensity as possible with as few peptide precursors as possible (Fig. 1a). It uses highly accurate predictions of fragment ion intensities and retention times for target and decoy peptides instead of spectral libraries.

In brief, predicted MS2 spectra from precursors with predicted retention times that fall within a data-dependent retention time window and precursor isotope envelopes that (partially) overlap with the isolation window are compared to experimental MS2 spectra. Matching is based on multiple fragment ion intensity-free and

-dependent scores for each PSM (Methods). Spurious PSMs are removed based on some of these scores. For example, PSMs are required to have at least three matched fragment ions, one of which must be the most abundant peak of the prediction and another one of which must be among the top three most-intense peaks of the prediction. PSMs passing these criteria are used for deconvolution, where they compete for experimental fragment ion intensity in one concerted step; an approach fundamentally different from classic methods (Fig. 1a). PSMs with enough contribution to the experimental spectrum as measured by CHIMERYS coefficients and that pass additional score filters are handed to mokapot[20] for PSM-level FDR control, specifically allowing for multiple PSMs per spectrum, similar to DIAmeter[27].

We validated this FDR estimation on data with varying chimericity by systematically increasing the isolation window width of 1-h single-shot measurements (pancreatic mouse cell digest) from 1.4 to 20.4 Th using entrapment experiments (Supplementary Methods). Figure 1b shows that CHIMERYS' peptide group-level $q$-values correspond to empirical $q$-values calculated based on entrapment identifications with the classic entrapment FDR (eFDR) approach, independent of isolation window width.

Figure 1c displays the confident identification of six precursors with relative contributions to the experimental total ion current ranging from 4% to 54% from a 2-h HeLa DDA single-shot measurement in a mirror spectrum. Their predicted retention times differ from the scan's observed retention time by 1.14 min on average, corresponding to less than 1% deviation relative to the gradient length of 120 min. Notably, the experimental intensities for the y1, y1-NH$_3$ and y1-H$_2$O ions that are shared between five of these precursors (C-terminal lysine) align well with the sum of their predicted intensities, scaled by their respective CHIMERYS coefficients, which can be interpreted as the interference-corrected total ion current of a precursor in an MS2 spectrum (Methods). This exemplifies how the algorithm identifies multiple peptides in chimeric spectra, while distributing intensities of shared fragment ions. Peptides identified by CHIMERYS recapitulate the expected quantitative ratios in a multi-organism-mixture experiment (Fig. 1d). This renders CHIMERYS suitable for approaches like wide-window DDA (also termed WWA or wwDDA)[28,29] and the analysis of DIA data.

To assess the performance of the algorithm on DDA data, we analyzed a 2-h HeLa cell digest with 1.3-Th MS2 isolation windows. CHIMERYS identified 238,795 PSMs at 1% run-specific PSM FDR with >85% of MS/MS spectra yielding one or more PSMs (identification rate; Extended Data Fig. 1a). More than two-thirds of the identified MS2 spectra contained more than one precursor (Extended Data Fig. 1b), confirming previous observations[6]. Fragment ions shared between different peptides were detected across the full MS2 $m/z$ range with an expected higher frequency ≤200 $m/z$ (Supplementary Fig. 1), rendering current strategies for handling chimeric spectra such as subtractive and multiplicative approaches error prone. Comparing these results to eight academic and commercial DDA search engines (Fig. 1e) revealed that CHIMERYS identifies many additional peptide groups (Extended Data Fig. 2a–c) in less time than was spent on data acquisition (Extended Data Fig. 2d). Most of these additional identifications were low abundant (Extended Data Fig. 3a). As such, they had fewer matched fragment ions than shared peptide groups (median of 10 versus 17; Extended Data Fig. 3b) but still high normalized spectral contrast angles[30] (median of 0.69 versus 0.85; Extended Data Fig. 3c and Supplementary Discussion). Hence, they are readily distinguished from decoys using mokapot's support vector machine score that aggregates CHIMERYS' score set (Extended Data Fig. 3d). Reassuringly, CHIMERYS-unique peptide groups markedly increased the number of peptides per protein group in CHIMERYS compared to Sequest HT (Extended Data Fig. 3e). It is worth noting that some of these search engines do not control FDR at the same level, which has a substantial influence on such comparisons (Extended Data Fig. 3f,g and Supplementary Table 1). Controlling FDR

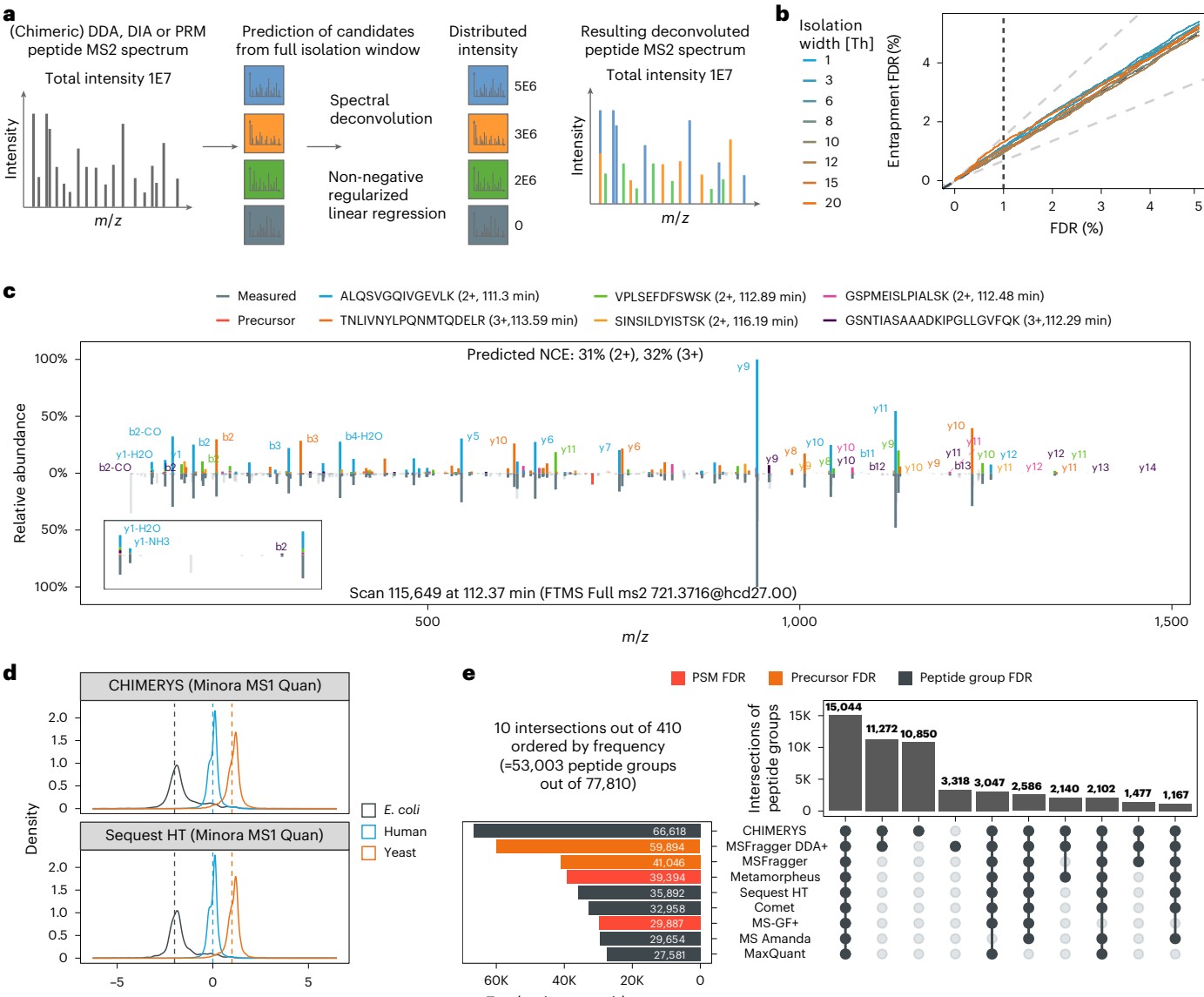

**Fig. 1 | Deconvolution of chimeric DDA spectra. a**, CHIMERYS treats chimeric spectra as linear combinations of pure spectra, and its spectrum-centric deconvolution uses non-negative regularized regression to estimate interference-corrected total ion currents for all candidate peptides. **b**, Peptide group-level entrapment analysis with the classic entrapment FDR (eFDR) approach (Supplementary Methods) of DDA data from a pancreatic mouse cell digest, acquired using different isolation window widths and processed with CHIMERYS. **c**, Example of a deconvoluted chimeric spectrum with six PSMs from a 2-h HeLa DDA single-shot measurement, acquired on an Orbitrap QE HF-X with 1.3-Th

isolation windows from the LFQbench-type dataset[37]. The inset visualizes how the experimental intensity of shared fragment ions with low *m/z* values is distributed to multiple PSMs. **d**, Peptide group-level log₂-ratio density plots for triplicate 2-h DDA single-shot measurements from two different conditions (*n* = 6), acquired on an Orbitrap QE HF-X with 1.3-Th isolation windows from the LFQbench-type dataset, analyzed with CHIMERYS (top) or Sequest HT (bottom). FDR was controlled at 1% at the global peptide group level. **e**, Comparison of peptide group identifications from multiple search engines on the same data as in **c**. FDR was natively controlled at different levels, depending on the search engine.

at a 'lower' level and counting identifications at a 'higher' level (for example counting peptides at PSM FDR) will usually overestimate the number of identifications. Identifications need to be reported at the same level at which FDR is controlled.

The gains observed for HeLa digests relative to Sequest HT (same protein grouping, as well as peptide group- and protein-level FDR estimation as CHIMERYS in Proteome Discoverer; PD) and MSFragger[31] (second highest number of identified peptide groups after CHIMERYS) were corroborated using CHIMERYS v.2.7.9 with more difficult biological samples at the protein group level (urine[32], +21%/+11%; CSF[32], +17%/+4%; plasma[32], +10%/−10%; formalin-fixed paraffin-embedded (FFPE) material, +35%/+21%; secretomes[33], between +33%/−4% and +71%/+27%, *Arabidopsis thaliana*[34], +13%/+1%; *Halobacterium*[34],

+20%/+6% for Sequest HT/MSFragger; Extended Data Fig. 4a–f), as well as using CHIMERYS v.4.0.21 with samples enriched for phosphorylated, acetylated and ubiquitinated peptides at the precursor level (phosphorylation[35], +64%/+36%; acetylation[36], +98%/+8%; ubiquitination[36], +88%/+45% for Sequest HT/MSFragger; Extended Data Fig. 4g). Extended Data Fig. 4h visualizes prediction accuracy of INFERYS v.4.0.0 for various post-translational modifications (PTMs). These data highlight that CHIMERYS substantially increases the analysis depth of DDA data.

## Revisiting legacy data using CHIMERYS

We conducted a retrospective study of HeLa single-shot analyses spanning many years and Orbitrap instrument generations. Despite many

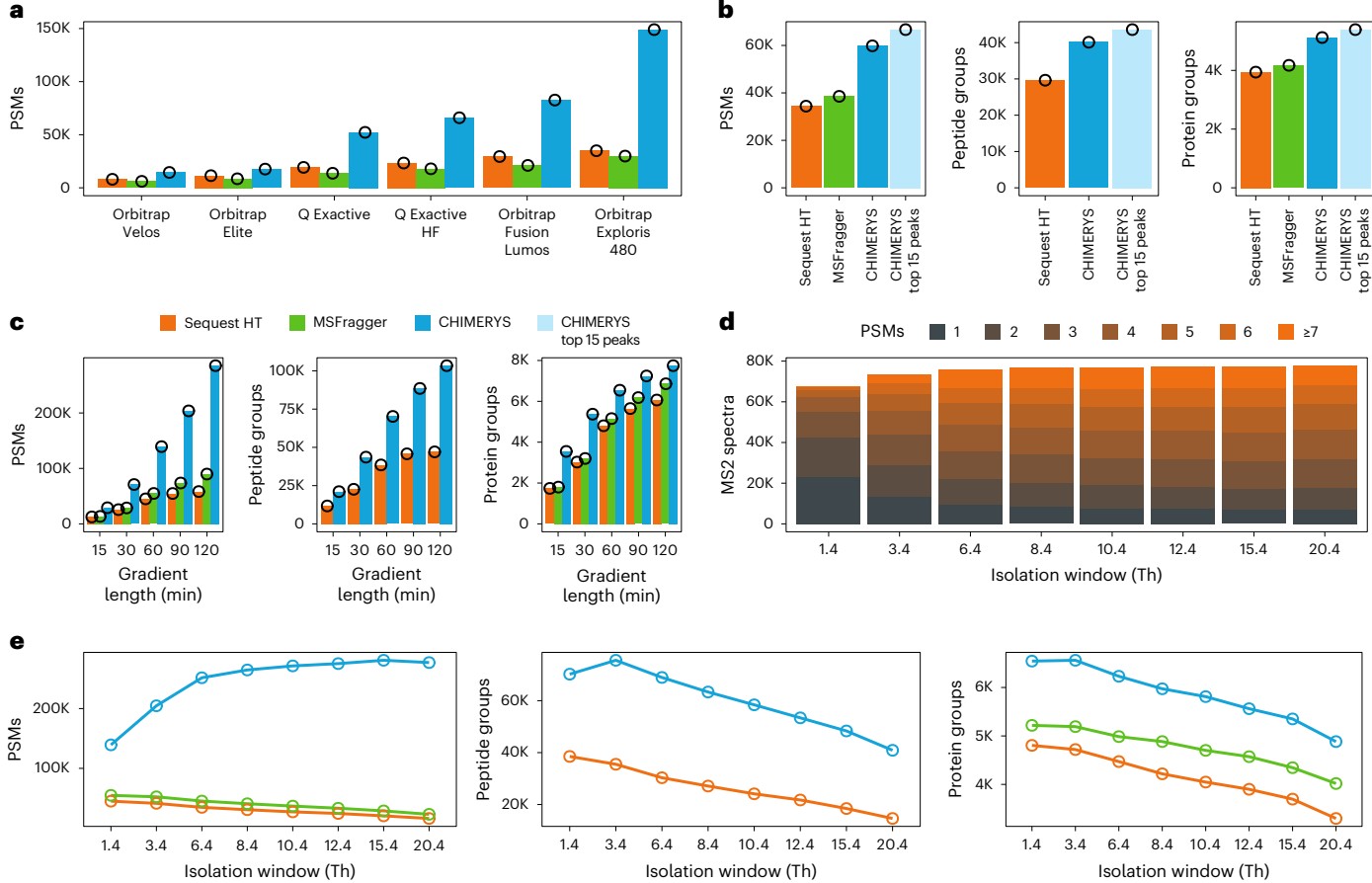

**Fig. 2 | Optimizing data acquisition with deconvolution in mind. a**, PSMs identified at 1% run-specific PSM-level FDR based on Sequest HT (orange), MSFragger (green) and CHIMERYS (blue) from 1-h HeLa single-shot measurements (*n* = 1), acquired using various Orbitrap generations. **b**, PSM, peptide group and protein group identifications based on Sequest HT (orange), MSFragger (green), CHIMERYS (blue) and CHIMERYS after removal of low-abundance peaks (light blue) from a 1-h HeLa single-shot measurement (*n* = 1), acquired using collision-induced dissociation (CID) fragmentation with ion trap readout. FDR was controlled at 1% at the run-specific PSM, peptide group (only available for Sequest HT and CHIMERYS) and protein group level, respectively. **c**, PSM, peptide group and protein group identifications based on Sequest HT (orange), MSFragger (green) and CHIMERYS (blue) from pancreatic mouse

cell single-shot measurements using various gradient lengths (*n* = 1). FDR was controlled at 1% at the run-specific PSM, peptide group (only available for Sequest HT and CHIMERYS) and protein group level, respectively. **d**, Distribution of the number of PSMs per MS2 spectrum from 1-h pancreatic mouse cell single-shot measurements, acquired using different isolation window widths (*n* = 1). FDR was controlled at 1% at the run-specific PSM level. **e**, PSM, peptide group and protein group identifications based on Sequest HT (orange), MSFragger (green) and CHIMERYS (blue) from 1-h pancreatic mouse cell single-shot measurements, acquired using different isolation window widths (*n* = 1). FDR was controlled at 1% at the run-specific PSM, peptide group (only available for Sequest HT and CHIMERYS) and protein group level, respectively.

differences that impair a fair comparison, a clear trend was observed, in that the higher the speed and sensitivity of the instrument, the higher the advantage of CHIMERYS over Sequest HT (Fig. 2a and Extended Data Fig. 5a).

Next, we investigated low-resolution ion trap data (ITMS), comparing CHIMERYS to Sequest HT on unprocessed spectra and on spectra filtered for the top 15 most abundant fragments per 100-Th window (Fig. 2b). In contrast to Orbitrap data, we observed a notable improvement by removing low-abundance peaks in ITMS spectra. Specifically, CHIMERYS identified 74% more PSMs, 35% more peptide groups, and 30% more protein groups compared to Sequest HT on unprocessed spectra, while it identified 94% more PSMs, 47% more peptide groups, and 37% more protein groups on spectra preprocessed with a top 15 by 100-Th filter from a HeLa digest. Both examples show that substantially more information can be extracted from legacy data by harnessing the information contained in chimeric spectra.

## Optimizing data acquisition with deconvolution in mind

We assessed to what extent CHIMERYS' capability to deconvolute highly complex spectra can be used to optimize data acquisition. First, we

evaluated LC gradients with the goal to increase sample throughput per day (SPD; Methods). Figure 2c shows that CHIMERYS identified a similar number of peptide and protein groups from a 30 min measurement of a pancreatic mouse cell digest (48 SPD) as Sequest HT from a 120 min measurement (12 SPD), increasing throughput by a factor of four.

Next, we explored a possible increase in identification efficiency by widening the isolation window in DDA (between 1.4 Th and 20.4 Th; Fig. 2d,e and Extended Data Fig. 5b–d). The analysis revealed that the number of identified PSMs from a pancreatic mouse cell digest increased with wider isolation windows and began to plateau at >8 *m/z*. In high-load samples like these, this is likely due to the automatic gain control (AGC) limit, which together with the dynamic range of MS2 spectra, limits the number of precursors in chimeric spectra with a sufficient number of detectable fragment ions. The number of unique peptide group (and protein) identifications reached its maximum already at a window size of 3.4 Th for this specific dataset and decreased for larger isolation windows. This is likely because more and more PSMs were from the same, high-abundant peptides that were now co-isolated more often. In contrast to that, it was previously shown for low-load samples that disabling the AGC limit together with extended injection

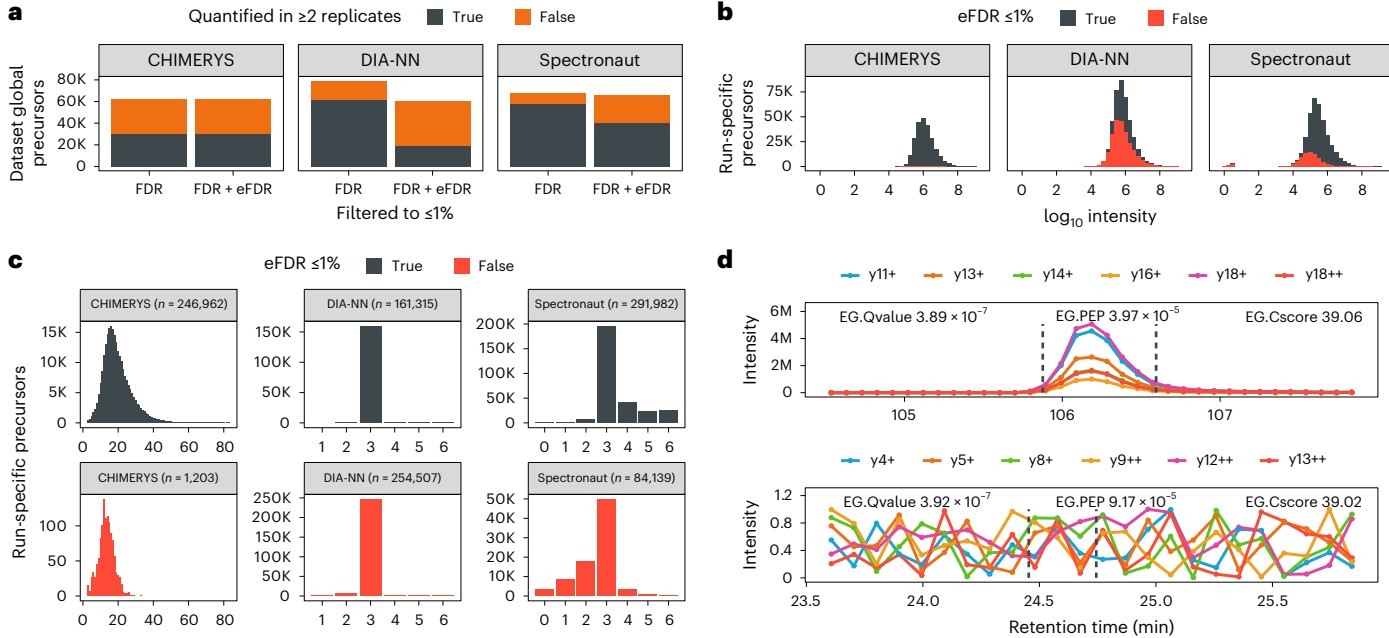

**Fig. 3 | Deconvolution of chimeric DIA spectra. a**, Precursors quantified by CHIMERYS, DIA-NN and Spectronaut in at least one (orange) or two (gray) out of three replicate 2-h DIA single-shot measurements of two conditions (*n* = 6). Data taken from the LFQbench-type dataset and acquired on an Orbitrap QE HF-X with 8-Th isolation windows. Identifications are filtered at 1% run-specific precursor-level FDR or additionally at 1% run-specific precursor-level eFDR (Supplementary Methods). **b**, Apex intensities for precursors surviving (gray) or not surviving (red) 1% run-specific precursor-level eFDR for the same data as in **a**, analyzed by CHIMERYS, DIA-NN and Spectronaut. **c**, Number of fragment ions used for the quantification of precursors by CHIMERYS, DIA-NN and Spectronaut for the same data as in **a**. Identifications are filtered at 1% run-specific precursor-level FDR and colored by whether they also survive 1% run-specific precursor-level eFDR (Supplementary Methods). **d**, Example fragment ion XICs exported from Spectronaut for two high-scoring precursors with similar confidence (all six library fragments are shown). The top one is also identified by CHIMERYS, while the bottom one is not. Spectronaut identifies some precursors based entirely on fragment ions with intensities below or equal to 1 according to their report. The bottom panel shows the highest-scoring one of these precursors. The vertical dashed gray lines mark Spectronaut's reported EG.StartRT and EG.EndRT for the corresponding precursors, with 105.88 min and 106.59 min for the top and 24.45 min and 24.74 min for the bottom XIC, respectively.

times enabled CHIMERYS to detect more unique peptides with very wide isolation windows[29]. There, the reduction in the number of MS2 scans due to the extended injection times resulted in a concomitant decrease in the number of peptide identifications with classic search engines, which CHIMERYS could counteract by identifying many PSMs from these highly chimeric, wide-window scans.

### Deconvolution of chimeric DIA spectra

CHIMERYS deconvolutes DIA spectra in the same way as DDA spectra. The only difference is that DIA spectra are usually more chimeric. Exemplified by a high-load LFQbench-type multi-organism mixture dataset[37], CHIMERYS identified an average of 529,993 PSMs per raw file at 1% run-specific PSM FDR, mapping to 66,888 unique peptide groups and 7,331 unique protein groups at 1% global peptide group and protein FDR, respectively, with an overall identification rate of >60% (Supplementary Fig. 2a). More than 82% of identified MS2 spectra contained more than one precursor (Supplementary Fig. 2b) and shared fragment ions were more frequent than in DDA, emphasizing the need for spectrum deconvolution that assigns shared fragment ions pro rata to the contributing peptides (Supplementary Fig. 2c–j).

### Comparison to other DIA search engines

We compared results obtained with CHIMERYS on DIA data to the library-free workflows implemented in DIA-NN[38] and Spectronaut[39] using entrapment experiments to validate FDR control in the run-specific context[40] (Supplementary Methods provide context definitions and search parameters). The results show that CHIMERYS' self-reported *q*-values correspond to empirical *q*-values calculated based on entrapment identifications (Extended Data Fig. 6a). DIA-NN

and Spectronaut seemed to underestimate FDR based on all three or the peptide and concatenated entrapment approaches, respectively (Extended Data Fig. 6b,c). Recently proposed more stringent settings for Spectronaut[41] had little, if any, effect on this issue (Extended Data Fig. 6d). Similar observations were made when analyzing the TimsTOF Pro data of the LFQbench-type dataset using DIA-NN and Spectronaut (Extended Data Fig. 6e,f). All subsequent analyses used the peptide eFDR approach. Using this approach, CHIMERYS v.4.0.21 finished the analysis of the dataset 4.9 times faster than DIA-NN and 1.7 times slower than Spectronaut (Extended Data Fig. 7a). Filtering at run-specific eFDR in addition to the algorithm-dependent self-reported FDR did not change the overall number of identifications (number of precursors identified in any number of replicates) for CHIMERYS, but it reduced the overall number of identifications for DIA-NN and Spectronaut to a level comparable to CHIMERYS. The number of precursors identified in two out of three replicates relative to the overall number of identifications (a measure for data completeness) did not change for CHIMERYS when filtering at run-specific eFDR in addition to the algorithm-dependent self-reported FDR (41% data completeness); however, doing so reduced data completeness for Spectronaut from 86% to 61% and for DIA-NN from 78% to 30% (Fig. 3a). CHIMERYS and Spectronaut substantially outperformed DIA-NN when requiring a precursor to be identified in all replicates at 1% algorithm-dependent self-reported FDR and at 1% run-specific eFDR (Extended Data Fig. 7b).

As expected, precursors filtered out based on eFDR have lower MS2 intensities (Fig. 3b) and fewer fragment ions (Fig. 3c); however, the extent to which this is observed differs substantially between the three tools. CHIMERYS considers more fragments for quantification than the other two search engines with default settings. Further, CHIMERYS

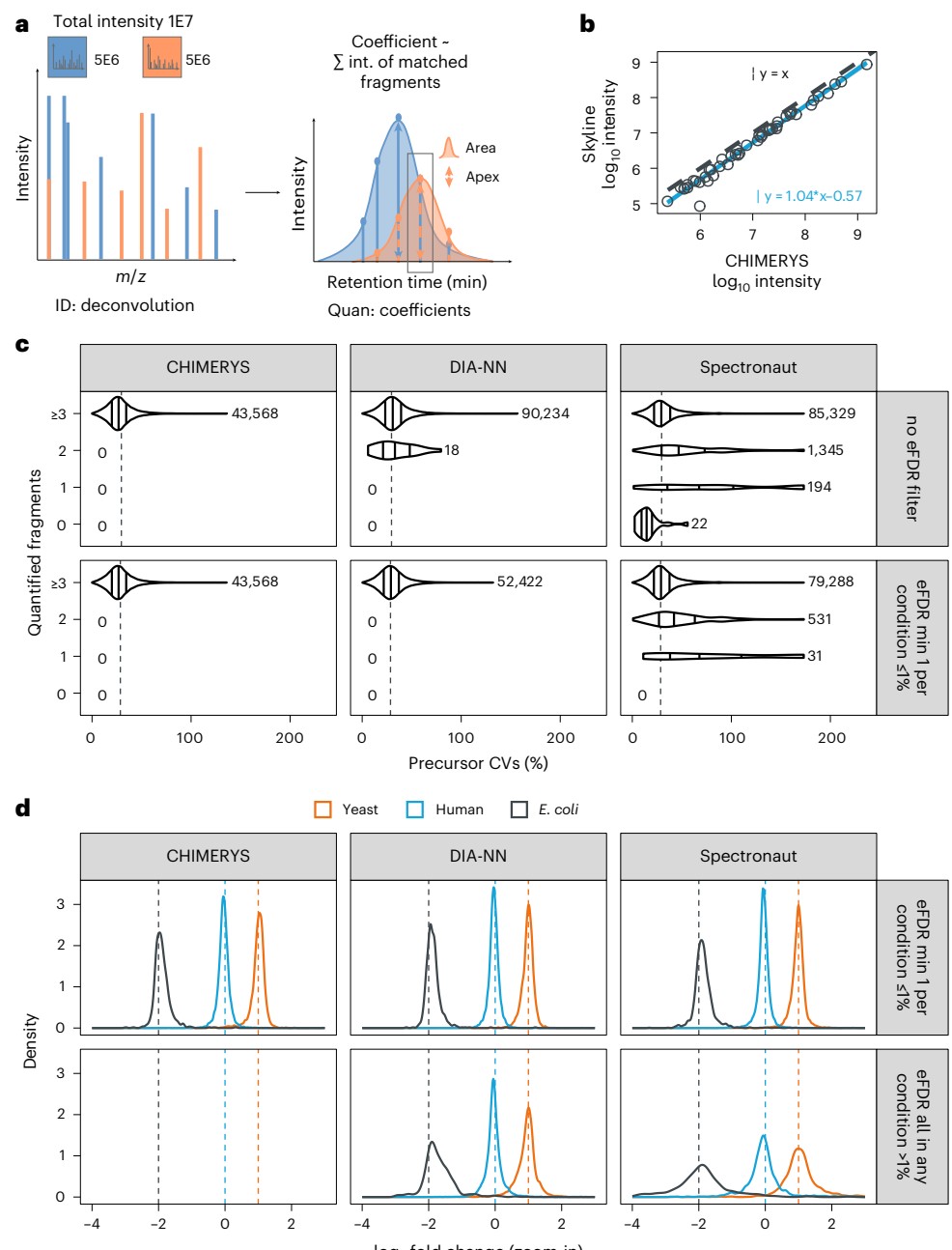

**Fig. 4 | Coefficient-based quantification. a**, CHIMERYS quantifies precursors based on MS2 spectra by tracing their spectrum-centric coefficients (interpretable as interference-corrected total ion currents) over retention time, followed by apex intensity extraction or trapezoidal peak area approximation. **b**, Scatter-plot of CHIMERYS' MS2-based quantification to Skyline's MS2-based quantification on a PRM dataset targeting 52 peptides from 18 human proteins across the entire dynamic range of a human proteome. The gray dashed line has a slope of 1 and the blue solid line is a linear fit to the data. **c**, Violin plots depicting CV distributions of precursors identified by CHIMERYS, DIA-NN or Spectronaut at 1% self-reported FDR (top) or additionally requiring at least one replicate per condition to also meet 1% entrapment FDR (bottom, peptide eFDR approach) at

the precursor level in the run-specific context, quantified based on 0, 1, 2 and 3 or more fragment ions. Data are from triplicate 2-h DIA single-shot measurements from two conditions ($n = 6$), taken from the LFQbench-type dataset and acquired on an Orbitrap QE HF-X with 8-Th isolation windows. Solid lines within violin plots mark the 1st, 2nd and 3rd quartile. Vertical dashed lines depict the median across all search engines with the respective filtering applied (29.6% for filtering at self-reported FDR and 28.9% with additional eFDR filtering). **d**, Precursor-level log₂-ratio density plots for the same data as in **c**, stratified by whether at least one replicate per condition survived 1% eFDR based on the peptide eFDR approach (top) or not (bottom). Replicate measurements were averaged before calculating ratios.

is more rigorous in the inclusion of fragment ions. The latter is illustrated in Fig. 3d. The top panel shows fragment ion chromatograms exported from Spectronaut for a precursor confidently identified by all three search engines. The bottom panel shows fragment ion chromatograms exported from Spectronaut for the highest-scoring, Spectronaut-unique precursor that was entirely based on fragment ions with an F.PeakArea ≤ 1. Inspection of the corresponding raw

data revealed that these fragment ions are missing in the relevant retention time range (Extended Data Fig. 7e and Supplementary Discussion). Both precursors were identified by Spectronaut with comparable scores and posterior error probabilities. Further investigations regarding the number and intensity of fragment ions (Extended Data Fig. 7c–e) suggest that precursors with less than three quantifiable fragment ions with an intensity exceeding 1 or those with (near-)zero

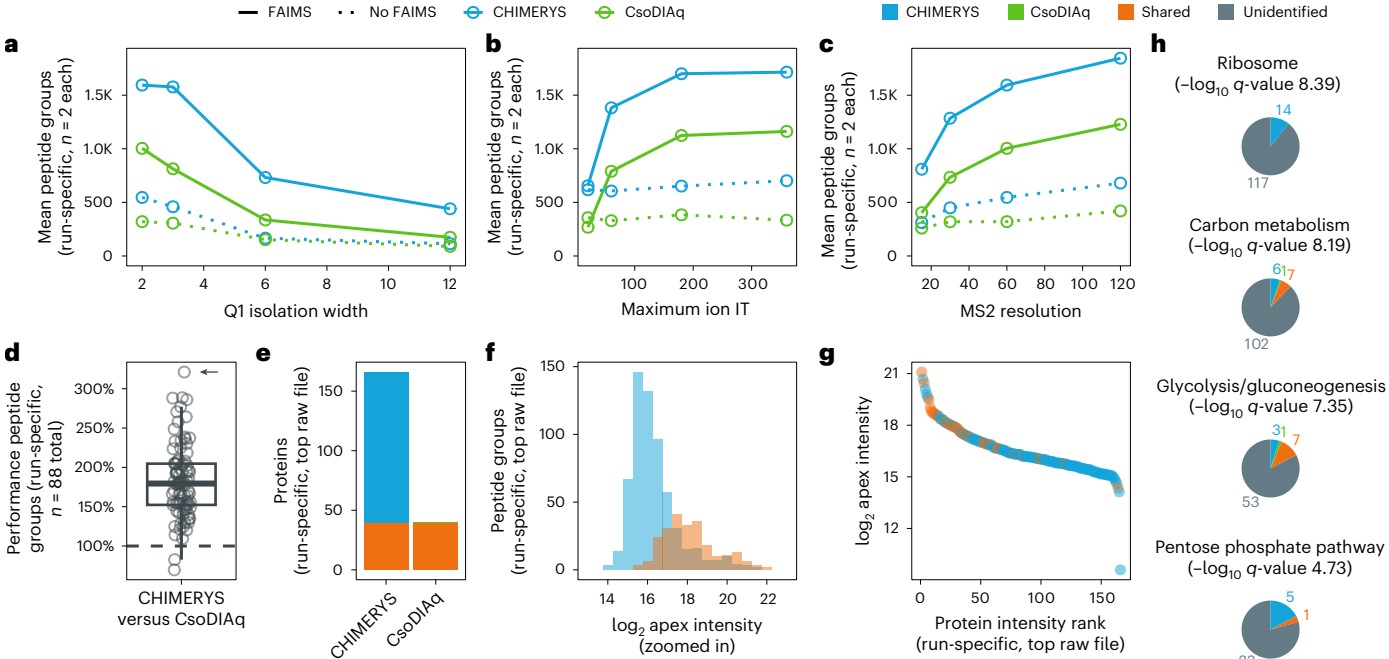

**Fig. 5 | Direct infusion data. a–c,** Mean number of peptide groups identified at 1% run-specific FDR by CHIMERYS in library-free DIA mode (blue) and the purpose-built search engine CsoDIAq[42] in spectral library mode (green) on a cohort of 88 DI-SPA MCF7 samples taken from Meyer et al.[7]. Dotted lines indicate samples measured without FAIMS[49]. Figures show a subset of samples split by Q1 isolation width (total *n* = 8 samples, measured in duplicate) (**a**), maximum ion injection time (total *n* = 8 samples, measured in duplicate) (**b**) and MS2 resolution (total *n* = 8 samples, measured in duplicate) (**c**). **d,** Boxplot visualizing the ratio of the number of peptide groups identified at 1% run-specific peptide group FDR by CHIMERYS versus CsoDIAq in each sample. The bottom (lower hinge), center and top of the box (upper hinge) are the first, second (median) and third quartile of the data points. The upper whisker extends from the hinge to the largest value no further than 1.5 × IQR from the hinge. IQR, or interquartile range, is the distance between the first and third quartiles. The lower whisker extends from the hinge to the smallest value at most 1.5 × IQR of the hinge. All individual data points are shown (*n* = 88 raw files). The arrow refers to the raw file referenced in **e–h. e–g,** Peptide groups and proteins identified at 1% run-specific FDR by both CHIMERYS and CsoDIAq (shared, orange), exclusively by CIMERYS (blue) or exclusively by CsoDIAq (green) in the sample marked with an arrow in **d.** Bar plot of identified proteins (**e**). Histogram of log₂-transformed peptide group intensity values (**f**). Rank plot of log₂-transformed protein intensity values (**g**). **h,** Cytoscape-based KEGG functional enrichment, showing the top 1, 2, 3 and 6 most statistically significant terms based on proteins identified by CHIMERYS. The number of proteins in each pathway are depicted as pie charts, colored by whether they were identified exclusively by CHIMERYS (blue) or CsoDIAq (green), by both (orange) or by neither (gray). All functional enrichment data are available in Supplementary Table 2 (Supplementary Methods).

intensity should be removed; either categorically or by applying stringent FDR control, which has a very similar effect (Extended Data Fig. 7f).

**Accurate peptide quantification from chimeric PRM and DIA spectra**

One of CHIMERYS' distinguishing concepts is its spectrum-centric processing of chimeric spectra. Apart from peptide identification, it also derives spectrum-centric quantitative information in the form of CHIMERYS coefficients, which can be interpreted as the interference-corrected total ion current for a given precursor in this MS2 spectrum (Methods). If none of the matched fragments for a precursor are shared with another precursor and the predicted MS2 spectrum matches perfectly to the experimental one, the coefficient is the sum of all matched fragment ions in the experimental MS2 spectrum. Hence, tracing the coefficient along retention time generates a pseudo-extracted-ion-chromatogram (XIC) that can be used to perform (relative) quantification of a precursor based on its MS2 signal in PRM and DIA data, but not in DDA data, which usually do not sample precursors multiple times along retention time in MS2. This is different from standard approaches that create XICs for (a subset of) fragment ions of a given precursor, which need to remove interfered fragment ions from quantification to maintain high precision and accuracy (Fig. 4a). To assess the performance of our concept, we carried out a simple PRM assay, focusing on 52 peptides from 18 human proteins spanning five orders of magnitude of cellular abundance (Methods). Both CHIMERYS and Skyline recovered 47 out of 52 peptides from the targeted inclusion list and CHIMERYS' automatically generated MS2-based quantification was in excellent agreement (Pearson correlation coefficient = 0.99) with the manually curated values obtained from Skyline (Fig. 4b). Without any additional effort, CHIMERYS identified and quantified 1,400 further peptides that were not designed to be in the assay but that happened to be co-isolated with the targeted peptides (Extended Data Fig. 8). CHIMERYS effectively automates the processing of PRM data because it removes the manual curation steps often required in Skyline. These include dealing with shared fragment ions and co-isolated peptides (both used in CHIMERYS but removed in Skyline).

Next, we compared the MS2-level quantitative precision and accuracy of CHIMERYS to DIA-NN and Spectronaut on the LFQbench-type dataset[37]. To avoid differences in quantification due to different methods for determining peak integration borders, we compared the three algorithms based on their implementation of peak apex quantification (Supplementary Methods provide DIA-NN- and Spectronaut-specific settings, as well as an explanation of the corresponding implementation in CHIMERYS). When filtering the data using eFDR as discussed above, the median quantitative precision of precursors (based on coefficient of variation; CV) was 26.9%, 29.1% and 29.2% for CHIMERYS, DIA-NN and Spectronaut, respectively (Fig. 4c).

Similarly, precursor-level ratio distributions (Fig. 4d) as a measure of quantitative accuracy for the three different search engines were comparable at eFDR (mean log₂ ratios ± s.d. for *Escherichia coli*, *Homo sapiens* and *Saccharomyces cerevisiae* of −1.90 ± 0.25, −0.03 ± 0.25 and 1.00 ± 0.29 for CHIMERYS, −1.86 ± 0.26, −0.03 ± 0.21

and 0.98 ± 0.23 for DIA-NN and −1.86 ± 0.32, −0.05 ± 0.31 and 1.00 ± 0.35 for Spectronaut, respectively). The above analysis demonstrates that CHIMERYS' spectrum-centric way of quantifying peptide precursors matches the performance of Skyline on PRM data as a gold standard in the field and extends to full-scale DIA data. It also highlights the potential of CHIMERYS for scaling PRM assays to very large numbers of peptides without the need for manual intervention.

## Digging deeper into direct infusion experiments

Recently, DI-SPA was shown to deliver proteomics insights at an unprecedented throughput[7]. As direct infusion experiments forfeit chromatographic separation in favor of sample throughput, such data are not readily accessible to algorithms such as DIA-NN or MSFragger-DIA. CHIMERYS natively supports the processing of DI-SPA data, because its spectrum-centric deconvolution of MS2 spectra does not require the detection and scoring of elution peaks in fragment ion XICs. As such, CHIMERYS is the only library-free algorithm capable of analyzing DI-SPA data. A reanalysis of the data underlying Fig. 2 of the original DI-SPA publication[7] with CsoDIAq[42], a tool designed for the analysis of DI-SPA data, and CHIMERYS revealed that the TraML library used by CsoDIAq contained a substantial number of decoys that differed in sequence length from their corresponding targets (Extended Data Fig. 9a,b). Generating hybrid and fully predicted spectral libraries with and without matched decoys (Supplementary Methods) revealed that this mismatch between targets and decoys artificially increased CsoDIAq's sensitivity (Extended Data Fig. 9c). Using matched decoys for both algorithms, CHIMERYS identified up to threefold more unique peptide groups in comparison to CsoDIAq at 1% run-specific peptide group FDR (Fig. 5a–d), resulting in up to threefold more identified proteins at 1% run-specific protein FDR (Fig. 5e). This is driven by ~twofold higher sensitivity of CHIMERYS compared to CsoDIAq (Fig. 5f,g). In turn, this leads to a substantial increase in significantly enriched KEGG[43] pathways (8 versus 17 for CsoDIAq and CHIMERYS; Supplementary Table 2) and their coverage with protein identifications (Fig. 5h). This demonstrates that CHIMERYS can unlock new biology hidden in previously acquired data that is inaccessible to other software solutions.

## Head-to-head comparison of DDA and DIA data, facilitated by CHIMERYS

We showed that CHIMERYS can analyze DDA and DIA data using the same concept for the deconvolution of chimeric spectra, which enables directly comparing the two acquisition methods on data acquired from the same sample, without the need to process the data with different software packages. As one would expect, it identified more than twice as many PSMs from DIA (8-Th isolation window) compared to DDA (1.3-Th isolation window) data acquired on an Orbitrap QE HF-X (LFQbench-type dataset; Extended Data Fig. 10a); however, DDA identified 52% more peptide groups and 30.3% more protein groups compared to DIA (Fig. 6a and Extended Data Fig. 10b). Likely, this is due to the interplay between the AGC limit and the dynamic range in MS2 spectra, which we already observed for WWA data (see section above). In contrast, relative quantitative data completeness was higher for DIA than for DDA data when filtering for peptide groups that met 1% FDR in the global, but not necessarily the run-specific context and enabling 'match between runs' for DDA using the Minora Feature Detector in PD[44] (78% versus 55.4% of peptide groups quantified in two out of three replicates per condition in DIA and DDA, respectively, Fig. 6a). This resulted in very similar numbers of peptide groups being quantified in two out of three replicates (56,322 and 52,161) for DDA and DIA, respectively.

Perhaps the more interesting comparison is that of DDA versus DIA using the same isolation window (here 2 Th). This has recently become possible because modern, fast-scanning instruments blur the border between DDA and DIA[45]. Interestingly, both 14 min

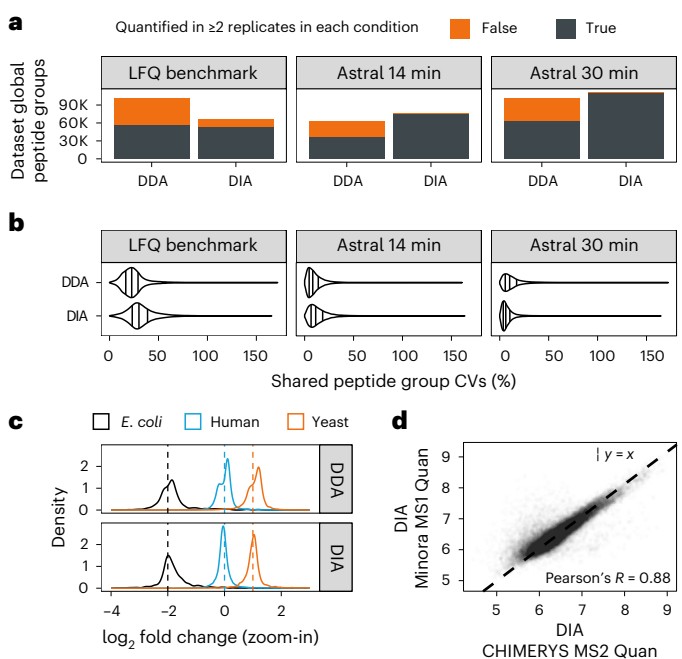

**Fig. 6 | Comparing DDA and DIA data. a**, Peptide groups identified in at least one (orange) or two (gray) out of three replicates by CHIMERYS in DDA or DIA data. For the LFQbench-type dataset, triplicate 2-h single-shot measurements from two conditions (n = 6) are shown. Data were acquired on an Orbitrap QE HF-X with 1.3- or 8-Th isolation windows for DDA and DIA, respectively. For the Orbitrap Astral[46] datasets, triplicate 14-min or 30-min single-shot measurements from a HeLa sample are shown. FDR was controlled at 1% at the global peptide group level. Match between runs was used for DDA data and for DIA data, peptide groups were quantified irrespective of their run-specific FDR. **b**, Peptide group-level CV distributions for shared peptide groups from the same data as in **a**. Solid lines within violin plots mark the 1st, 2nd and 3rd quartile. **c**, Peptide group-level log2-ratio density plots for shared peptide groups from the same data from the LFQbench dataset as in **a**. **d**, Scatter-plot of peptide group-level apex quantification between CHIMERYS' MS2-based quantification (x axis) and Minora's MS1-based quantification (y axis) for a DIA raw file from the LFQbench dataset (n = 1). Data points are semitransparent. The Pearson correlation coefficient is included in the plot. The black dashed line has a slope of 1.

(~100 SPD) and 30 min (48 SPD) gradients on an Orbitrap Astral[46] yielded similar numbers of PSMs, peptide and protein groups for DDA and DIA (Fig. 6a and Extended Data Fig. 10a,b). The small differences in favor of DIA are likely due to the higher scan rate of the Orbitrap Astral in DIA mode. Again, relative quantitative data completeness was much better for DIA than for DDA (97.9% and 98.7% of peptide groups quantified in two out of three replicates versus 56.3% and 61.7% for the 14-min and 30-min gradients, respectively; Fig. 6a). These data suggest that DIA and MS2-based quantification should be preferred over DDA and MS1-based quantification when performing label-free, single-shot measurements on fast-scanning instruments. Comparing CV distributions of peptide groups detected by DDA and DIA in the three datasets revealed that DDA was slightly more precise on the LFQbench-type dataset, whereas DIA was slightly more precise on the 30 min Orbitrap Astral dataset (Fig. 6b). Quantitative accuracy seemed to be generally better for DIA on the LFQbench-type dataset (Fig. 6c); however, closer inspection suggests that this is due to a problem with the samples rather than with MS1-based quantification per se, as the accuracy of MS1- and MS2-based quantification of the DIA data is comparable (Extended Data Fig. 10c). In fact, CHIMERYS' MS2-based quantification was highly correlated (R = 0.88) to the MS1-based quantification implemented in PD on the same raw data (Fig. 6d), suggesting that the two quantification methods could be combined in the future in CHIMERYS.

## Discussion

In many ways CHIMERYS returns to the very old concept of analyzing tandem mass spectra, one at a time. At least for the task of peptide identification, this so-called spectrum-centric approach places the core analytical evidence acquired by the mass spectrometer at the center of data analysis. This comes with a number of important advantages. First, any proteomic data type (DDA, DIA and PRM) can be treated the same, and CHIMERYS is the first software implementation that stringently follows this unifying philosophy. While some tools such as MSFragger[16] and MaxQuant[47] claim to unify DDA and DIA data analysis, these are not unifying algorithms but bundles of acquisition method-specific algorithms with a unified user interface. In contrast to that, a unifying algorithm natively supports the analysis of different data types. Second, in principle, there is no difference between identifying a single or multiple peptides from the same MS2 spectrum and skilled scientists have done so since the early days of proteomics. The added sophistication is that artificial intelligence can predict the fragmentation of any peptide with outstanding accuracy so that it is possible to deconvolute even highly chimeric spectra by maximizing the explained intensity in an MS2 spectrum using a minimal set of peptides. Third, statistical methods for PSM-level FDR control are conceptually well worked out and have reached a very high level of practical refinement, again including the use of artificial intelligence that can predict the tandem mass spectrum of any target or decoy peptide with the same accuracy, ensuring fair competition between targets and decoys. Fourth, the plausibility of an identification can be further assessed (albeit not automatically) beyond statistics by visual inspection in the context of the full MS2 spectrum and for example by looking for fragment ions that were not part of the deep-learning model and have thus not yet been used for identification. A current limitation of CHIMERYS in this context is that peptides carrying modifications that are not yet covered by the underlying deep-learning model escape detection. It can be anticipated that this limitation will diminish over time as deep-learning models start to emerge that are capable of generalizing to modifications or fragmentation methods they have not yet been trained for[48] (Supplementary Discussion provides a more in-depth look on extrapolation). Similarly, CHIMERYS is currently limited to the analysis of data generated by mass spectrometers from Thermo Fisher Scientific; however, support for mzML and other vendor-specific formats will be available in a future version of CHIMERYS.

Akin to other software tools, CHIMERYS also uses the information contained in the MS2 spectrum for peptide quantification; however, unlike all other DIA software, it does not set a fixed number of fragment ions to consider and instead always uses all the fragment ions that have led to an identification in a given MS2 spectrum, but in relative proportion to how much they contributed to the actual signal in the MS2 spectrum (important for the frequent case of fragment ions that are shared between peptide candidates). CHIMERYS uses the sum of these fragment ion intensities rather than the individual fragment intensities to find the apex of a chromatographic peak. This makes the overall quantification more robust against weak signals and spurious detections as encountered for example in single-cell proteomics data. The results indicate that quantitative precision and accuracy closely match that of PRM data, which are often considered to be the gold standard for peptide quantification. In this context it is interesting to note that CHIMERYS also automates the analysis of PRM experiments along the way.

We consciously decided to rate data quality over quantity, such that reported peptide identification and quantification results are rather conservative and other software tools may sometimes seemingly outperform CHIMERYS (Supplementary Discussion); however, when applying rigorous and consistent criteria for peptide detection and quantification, these differences diminish. A perhaps unexpected finding in this regard is that DIA data are often not nearly as complete

as default processing parameters of DIA search engines report. Again, and not surprisingly, this is particularly true for low-abundant samples or low-abundant peptides within a sample. The reasons for this could be manifold and investigating them comprehensively goes beyond the scope of the present study; however, it is worth mentioning that the most recent generation of mass spectrometers has driven sensitivity to the point of single ion detection. As a result, MS2 spectra have at least some low level of signal at nearly every $m/z$. Many of these may not even stem from peptides but will create a situation in which 'something' can be found everywhere and all the time, potentially leading to data completeness that bears little if any actual justification. In addition, the increasing volume and density of MS-based proteomic data keeps challenging the scalability of the assumptions underlying data-processing tools. Reassuringly, the community of proteomics software developers and users are increasingly aware of these recurring challenges, as it is in everybody's best interest to ensure that software tools can be trusted and used at face value. CHIMERYS makes a valuable contribution in this context and a particularly exciting prospect is that the latest LC–MS/MS hardware along with the latest software solutions will soon overcome the historically grown divide in the field between DDA and DIA.

## Online content

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

## Methods

### Brief description of the CHIMERYS algorithm

**Setup.** The CHIMERYS workflow is a cloud-native web service with an application programming interface, orchestrated by Kubernetes on Amazon Web Services or on-premise. The environment consists of two major components: an INFERYS prediction server[25], which delivers predictions via remote procedure calls to a CHIMERYS search algorithm instance, which matches these predictions to experimental spectra. We decided to set up a cluster of servers for predictions, rather than allowing them to be performed locally to reduce the runtime of CHIMERYS (Supplementary Discussion).

**Description of the identification workflow.** The CHIMERYS workflow follows the setup of classic search engines. After the in silico digest of the protein database and the generation of shuffled decoy sequences using a similar logic as the mimic[21] entrapment generator (see section below), a coarse first search is performed to identify highly confident peptides for recalibration purposes. Notably, for a group of I/L isomers, CHIMERYS only scores one representative. A fast fragment ion index implementation similar to MSFragger[31] is used to determine a ranked list of suitable candidate peptides with isotope envelopes that (partially) overlap with the MS2 isolation window (plus tolerances). Fragment ion intensities for highly ranking candidate peptides are predicted for each spectrum, merged against the experimental MS2 spectrum and subsequently, a set of counting-based (for example, number of matching peaks between predicted and experimental spectrum) and intensity-based scores (for example, normalized spectral contrast angle) are calculated. Candidate peptides that fall below certain cutoff criteria are removed and only the best-scoring PSM for a group of isobaric PSMs is retained. For example, PSMs are required to have at least three matched fragment ions, one of which must be the base peak (most abundant peak of the prediction) and another one of which must be among the top three most-intense peaks of the predicted spectrum. After the initial search, a linear discriminant analysis identifies highly confident PSMs for the calibration of optimal prediction parameters of the fragmentation model (for example, normalized collision energy; NCE), refinement learning of the retention time model and recalibration of fragment ion $m/z$ and match tolerances. Peptide classes with few confidently identified peptides are removed entirely from the search space (for example, peptides of length 7 carrying two missed cleavages and two oxidized methionine residues). In the main search, the above-described scoring functions are executed using the optimized settings and prediction parameters, albeit now also filtering candidate peptides based on their predicted retention time. CHIMERYS uses retention time tolerance windows that would allow the identification of all peptides confidently identified in the initial search. In brief, we calculate the absolute difference of experimental and predicted retention times after refinement learning for all PSMs identified at 1% FDR in the initial search. The 100% quantile of these differences is the basis for the initial retention time window (±100% quantile of absolute differences between experimental and predicted retention time), which is then expanded further by multiplying it with a security factor of 2.5. The scoring is repeated to arrive at a set of high-scoring candidate peptides as input for the deconvolution function, where the candidate peptides simultaneously compete for experimental fragment ion intensity in one concerted step.

Essentially, CHIMERYS treats chimeric spectra as linear combinations of pure spectra. Here, predicted spectra for high-scoring candidate peptides serve as the source of pure spectra, which is why CHIMERYS is dependent on accurate fragment ion intensity predictions. The intensities of each predicted spectrum are normalized to a total sum of 1. Let $P$ be a collection of predicted spectra for high-scoring candidate peptides, each comprising a set of intensities $I_{p,m}$, where $p = 1, 2, \dots, P$ is an index over the predicted spectra and $m = 1, 2, \dots, M$ is an index over mass channels. The mass channels are defined by the

peaks of the experimental spectrum that were within the recalibrated fragment ion match tolerance of at least one fragment ion from the collection of predicted spectra, as well as all unmatched fragment ions from said collection. As such, if two peaks from two different predicted spectra match to the same experimental peak, they will have the same mass channel. This is how CHIMERYS handles shared fragment ions. Hence, one can represent the predicted spectra of all candidate peptides as a $P x M$ matrix of intensities $I_{p,m}$. The CHIMERYS coefficients $\beta_p$ then define a combined spectrum as the linear combination of the predicted spectra, scaled by the corresponding coefficients. This combined spectrum has a set of intensities:

$$\mathbf{I}_m = \sum_{p=1}^{P} \beta_p \mathbf{I}_{p,m}$$

CHIMERYS uses non-negative L1-regularized regression via the LASSO to optimize the CHIMERYS coefficients $\beta_p$ such that they minimize the following objective function:

$$\sum_{m}^{M} (I_m^{\exp} - \mathbf{I}_m^T \boldsymbol{\beta})^2 + \lambda ||\boldsymbol{\beta}||_1$$

subject to the coefficients $\beta_p$ having non-negative values. Here, $M$ is the number of mass channels, $I_m^{\exp}$ is the experimental intensity for the mass channel $m$, $\mathbf{I}_m = (I_{1,m}, I_{2,m}, I_{3,m}, \dots, I_{P,m})$ is a vector of predicted intensities for each predicted spectrum from the collection at mass channel $m$, $\boldsymbol{\beta} = (\beta_1, \beta_2, \beta_3, \dots, \beta_p)$ is a vector of CHIMERYS coefficients and $\lambda ||\boldsymbol{\beta}||_1 = \lambda \sum_p^P |\beta_p|$ is the LASSO regularization term (L1-regularization). $\lambda$ may be varied to vary the strength of the regularization, and therefore the strength of the constraint on the number of nonzero coefficients. CHIMERYS optimizes $\lambda$ automatically by fitting multiple models with different regularization strengths and selecting the best model with the most regularization by inspecting the corrected Akaike information criterion. As such, CHIMERYS models the experimental spectrum as a function of the matrix of candidate peptides. By using L1-regularization together with the corrected Akaike information criterion, it aims to best explain the experimental spectrum with the fewest number of candidate peptides possible. Notably, this algorithm accounts for the presence of shared fragment ions. The above-mentioned optimization procedure will effectively 'distribute' their intensity to the corresponding candidate peptides according to the optimized CHIMERYS coefficients. The more similar the sequences of two co-eluting peptides are, the more fragments they share and the more similar their predicted spectra will be. This is particularly true for positional isomers of PTM-containing peptides that often share many fragment ions. However, also unmodified peptides with the same amino acid composition can have very similar, but also completely dissimilar sequences and hence predicted spectra. The calculation of CHIMERYS coefficients is the same, no matter how many fragment ions are shared between two co-eluting peptides. The only exception are isobaric peptides. Currently, if two candidate peptides are isobaric, CHIMERYS will only insert the best one for a given spectrum into the collection of predicted spectra mentioned above. This is similar to MS1-based quantification, where only the best-scoring peptide is matched to an MS1 signal based on precursor $m/z$ and retention time. However, positional isomers usually have some site-determining ions that are isomer-specific and, depending on the PTM and its localization, fragment ion intensities might also differ between them. Hence, in the future, CHIMERYS might report multiple isomers per scan if their predicted spectra are sufficiently different from one another such that the LASSO regression assigns both of them a non-negative CHIMERYS coefficient.

The CHIMERYS coefficient for each candidate peptide represents its contribution to the experimental spectrum. A coefficient >0 indicates that this candidate peptide was used to explain the experimental

spectrum. Based on the resulting coefficient, a subsequent round of intensity-based scoring is executed. Here, the coefficients of the candidate peptides can be used to predict the proportional intensity of all but one candidate peptide, add them together and subsequently subtract this sum from the actual experimental MS2 spectrum to calculate what we call a 'shadow spectrum', which is the experimental spectrum with the contributions of all interfering peptides removed. Next, the above-mentioned figures of merit are calculated based on these shadow spectra without the interference of other peaks in the spectrum. Notably, this also works for fragment ions that are shared between candidate peptides. Candidate peptides that fail to meet certain quality criteria (for example, minimum number of most abundant peaks shared between predicted and experimental spectrum) are filtered out. A list of all remaining target and decoy PSMs per spectrum that received a coefficient >0 and met all quality criteria including all calculated scores is generated as input for the PSM-level error estimation in mokapot[20].

**FDR estimation using mokapot.** For error control, the initial implementation of CHIMERYS utilized Percolator[21] v.3.0.5 to aggregate all calculated scores for all target and decoy PSMs generated in a dataset. As Percolator runtime scales poorly with large input files, we exchanged it with the Python-based reimplementation termed mokapot[20]. To ensure scalability to large input PSM lists while controlling the compute resources, we rewrote large parts of mokapot's logic to allow streaming of data from disk, introduced RAM limits and implemented more-performant data structures. The changes made to mokapot have since been merged into the main branch (https://github.com/wfondrie/mokapot). Mokapot is executed using the following parameters: Training FDR of 1%, a training subset of 400k and ten iterations for training. We specifically prevent mokapot from only retaining the top-scoring PSM per spectrum. Afterwards, the resulting PSM-level $q$-values, support vector machine scores and posterior error probabilities are attached to the corresponding PSMs. Peptides containing leucine/isoleucine (I/L) isomers in the search space are added back to the results with identical scores and are flagged as 'ambiguous'.

**MS2 quantification workflow.** CHIMERYS determines raw file-specific peak apex retention times as the CHIMERYS coefficient-weighted mean of retention time deltas relative to the gradient length based on PSMs meeting 1% run-specific PSM-level FDR. If an external inclusion file was used, PSMs meeting 1% run-specific PSM-level FDR including their relative retention times and CHIMERYS coefficients from the list are also considered. If no PSMs meet 1% run-specific PSM-level FDR for a given precursor in a given raw file, the apex for said precursor in this raw file is calculated using the same logic as above but based on PSMs meeting 1% run-specific PSM-level FDR in other raw files and the inclusion file. CHIMERYS in its current implementation then estimates maximum integration borders per raw file as the 99% quantile of peak widths at base (not full width at half maximum) from precursors with at least three PSMs surviving a run-specific PSM-level FDR threshold of 1%. These maximum integration borders are then applied to each precursor in this raw file, leading to relatively wide integration borders, particularly for low-abundant precursors. Afterwards, quantification of PRM and DIA data is performed by either trapezoidal integration of the CHIMERYS coefficients from each precursor in a set of consecutive MS2 spectra sharing the same isolation window within the integration borders, or by using the highest CHIMERYS coefficient within the integration borders as the elution peak apex intensity. The latter implementation was used for the comparison to DIA-NN and Spectronaut. One missing CHIMERYS coefficient in a series of consecutive MS2 scans with the same isolation window is allowed (gap scan) and a contribution of 0 is inserted to any further scan with missing data points, which act as boundaries for peak area integration. Notably, at this point, CHIMERYS coefficients are taken from PSMs irrespective

of their run-specific PSM-level FDR; however, CHIMERYS coefficients will only be used from peptide precursors that meet CHIMERYS' quality criteria (for example, a minimum of three peaks matched between the predicted and the experimental spectrum) and are located in the vicinity of the determined peak apex. Hence, at least one confidently identified PSM across all raw files is required to generate quantitative values based on PSMs around the determined peak apex in each raw file. As such, CHIMERYS will quantify precursors that fail to meet run-specific precursor-level FDR thresholds. Users are free to filter their list of precursors at 1% global precursor-level FDR (precursor was confidently identified in at least one raw file) or additionally also at 1% run-specific precursor-level FDR. The latter will reduce data completeness and is more conservative; however, we have shown that often, these quantifications are precise and accurate, so we recommend to work with precursors filtered to 1% global precursor-level FDR during exploratory data analysis and turn to run-specific precursor-level FDR for the validation of interesting hits.

**Post-processing of CHIMERYS' PSM-level outputs.** CHIMERYS v.2.7.9 as showcased in this study is integrated into Thermo Fisher Scientific PD software v.3.1.0.622 (PD)[44]. A pre-release of PD v.3.2 was used to demonstrate the processing of PTM datasets with CHIMERYS v.4.0.21 (Extended Data Fig. 4g). Hence, PD starts CHIMERYS searches on Amazon Web Services by uploading an internal format containing only MS2 spectra and some auxiliary information, a fasta file and the search parameters to the CHIMERYS web service, which then processes the data and generates a result file. The result file is then downloaded and post-processed by PD[44]. In this study, we used the default CHIMERYS processing and consensus workflows with minor modifications. In brief, all DDA data processing was carried out using the PSM Grouper node to generate peptide groups, which were then validated using the Peptide Validator node. For DIA data, we used a special PCM Grouper node, which enables the calculation of run-specific and global precursor-level FDR. MS1-based quantification was performed using the Minora Feature Detector with default settings. MS2-based quantification was performed using the MS2 Fragment Ions Quantifier node with default settings.

## Data generation

**Cell culture and sample preparation.** Human HeLa (ATCC, CCL-2) and pancreatic mouse cells were cultured under standard conditions at 37 °C with 5% $CO_2$ in DMEM supplemented with 10% fetal bovine serum and 100 U ml$^{-1}$ penicillin (Invitrogen). At around 80% confluence, cells were washed three times with PBS buffer before urea lysis (8 M urea, 80 mM Tris, pH 7.6 and 1× protease inhibitor) was performed for 5 min on ice. Cell lysate was clarified by centrifugation (20,000$g$ for 10 min).

In-solution protein digestion was conducted as follows. First, proteins were reduced with 10 mM dithiothreitol at 37 °C for 1 h, followed by alkylation with 2-chloroacetamide at a final concentration of 55 mM for 45 min at room temperature in the dark while shaking on a thermo shaker. After the addition of five volumes of 50 mM Tris (pH 8), trypsin digestion was performed overnight by adding trypsin twice (1:100 dilution) after a primary incubation time of 4 h. Desalting was performed using Sep-Pak columns according to the user manual. Human brain FFPE samples were digested using an SDS lysis protocol followed by digestion with the SP3 approach as described in detail by Tüshaus et al.[33].

**LC–MS/MS.** FFPE, gradient comparison and wwDDA data were acquired on a micro-flow LC coupled via a HESI source to an Q Exactive HF-X hybrid quadrupole-Orbitrap mass spectrometer (Thermo Scientific). Optimization of the micro-flow LC setup as well as technical details were previously published by Bian et al.[32]. In brief, peptide separation was performed on an Acclaim PepMap 100 C18-HPLC-column (15-cm length, 1-mm inner diameter, 2-μm particle size; 164711, Thermo Fisher

Scientific) at 55 °C. Linear gradients with buffer A (0.1% v/v formic acid (FA) and 3% v/v dimethylsulfoxide (DMSO) in dH₂O) and buffer B (0.1% v/v FA and 3% v/v DMSO in acetonitrile) from 3% to 28% B were run at 50 μl min⁻¹. Sample loading, column wash and equilibration was performed at 100 μl min⁻¹. Source settings were applied as 320 °C capillary temperature, 3.5 kV spray voltage and 300 °C auxiliary gas. MS data were acquired at a normalized collision energy of 28%, in Top20 mode, at an *m/z* range of 360–1,300, AGC target of 3E6 and 1E5, maximal injection time of 50 ms and 22 ms, resolution of 60 k and 15 k on MS1 and MS2 level, respectively. The MS2 isolation window width was 1.4 Th in standard DDA runs and increased up to 20.4 Th for wide-window acquisition DDA as indicated in the figure legends.

Ion trap data were acquired with an Orbitrap Eclipse Tribrid mass spectrometer (Thermo Scientific) that was coupled to a Dionex UltiMate 3000 RSLCnano System (Thermo Scientific). Samples were transferred onto a trap column (75 μm × 2 cm, 5 μm C18 resin Reprosil PUR AQ; Dr Maisch). After washing with the trap washing solvent (5 μl min⁻¹, 10 min), samples were separated on an analytical column (75 μm × 48 cm, 3 μm C18 resin Reprosil PUR AQ, Dr Maisch). A 70-min method, including a 50-min gradient, was performed with a flow rate of 300 nl min⁻¹ (4% B up to 32% B within 50 min). Solvent A was 0.1% v/v FA and 5% v/v DMSO in dH₂O; solvent B was 0.1% v/v FA and 5% v/v DMSO in acetonitrile. MS1 scans were acquired with an Orbitrap resolution of 60 k, within a scan range of 360–1,300 *m/z*, a maximum injection time of 50 ms, a normalized AGC target of 100% and RF lens of 40%, including charge states 2–6, with an exclusion time of 25 s. MS2 scans were performed with the ion trap with a normalized AGC target of 200%, a maximum injection time of 25 ms and either with a higher-energy collisional dissociation (HCD) collision energy of 31% (wwDDA) or with a CID collision energy of 35% (CID). The quadrupole isolation window varied between 0.4, up to 5.0 *m/z* as indicated in the figure.

Data for the instrument comparison were assembled from 1-h HeLa quality control runs, acquired over several years at the Chair of Proteomics and Bioanalytics at the Technical University of Munich. They were run on various LC systems, employed diverse instrument-specific settings, slightly different gradients and used different batches of HeLa digest, prepared in house.

**Targeted assay generation.** A simple PRM assay was devised by randomly selecting 18 proteins and 2–3 peptides each across the whole measured intensity range from a 1-h HeLa run analyzed on an Orbitrap Fusion Lumos mass spectrometer (Thermo Scientific). A total of 51 precursors were put into an inclusion list in addition to 14 precursors corresponding to a retention time standard. A 1-h HeLa sample was analyzed in PRM mode: MS2 spectra were acquired using 0.4-Th isolation window, a maximum injection time of 100 ms, HCD collision at a normalized collision energy of 28% and readout in the Orbitrap at 15 k resolution.

### Reporting summary
Further information on research design is available in the Nature Portfolio Reporting Summary linked to this article.

### Data availability
The following external data were downloaded from PRIDE or MassIVE and processed with the respective search engines. In brief, body fluid data from Bian et al. (PXD015087)[32], secretome data from Tüshaus et al. (PXD018171)[33], *Arabidopsis thaliana* and *Halobacterium* data from Müller et al. (PXD014877)[34], phosphorylation data from Frejno et al. (PXD013615)[35], acetylation and ubiquitination data from Zecha et al. (PXD023218)[36], triple species mix as well as HeLa data from the LFQBench-type dataset by Van Puyvelde et al. (PXD028735)[37], Orbitrap Astral data extracted from Gutzman et al. (PXD046453)[46] and DI-SPA data from Meyer et al. (MSV000085156)[7]. Notably, peptides containing methionine residues were excluded from all analyses of the LFQBench-type dataset, as raw files might show differential oxidation. The same applies to the Orbitrap Astral data from PXD046453. For the LFQBench-type dataset, technical replicates were analyzed (Supplementary Table 3). All other replicates are biological replicates. An itemized mapping of external data processed as part of this study to their source is available in Supplementary Table 3. The following datasets were generated in house: FFPE (biological replicates), gradient comparison, wwDDA, instrument generations and PRM data. An overview of the files generated is provided in Supplementary Table 4. The generated MS raw and search data of internal datasets from this study are available via PRIDE[50] with the dataset identifier PXD053241. All fasta files used in this study are available via PRIDE[50] with the dataset identifier PXD053241. All Source and Supplementary Data files required to reproduce this study are available via PRIDE[50] with the dataset identifier PXD053241.

### Code availability
The mokapot version used in this study is available on GitHub (https://github.com/wfondrie/mokapot/). The modifications to the mimic entrapment database generator are available on GitHub (https://github.com/percolator/mimic/). A web version of the mimic tool can be found at https://mimic.msaid.io/. A demo version of PD and CHIMERYS can be requested at https://www.msaid.de/chimerys-demo or by contacting the corresponding authors. The custom R scripts used for data analysis are available on GitHub (https://github.com/msaid-de/chimerys-manuscript).

### References
50. Perez-Riverol, Y. et al. The PRIDE database resources in 2022: a hub for mass spectrometry-based proteomics evidences. *Nucleic Acids Res.* **50**, D543–D552 (2021).

### Acknowledgements
The authors thank numerous scientific colleagues for their input, discussions and support. The authors expressively thank the PD software development team at Thermo Fisher Scientific for their collaboration, support and contributions to the successful integration of CHIMERYS into PD and the scientific discourse on the results. The authors thank E. Zander for consulting on mathematical topics. The authors thank M. The for consulting on entrapment experiments and FDR control. The authors also thank their colleagues D. Bold, J. Santoso and A. Guevende for contributions to the software. This work was partially supported by a European Research Council Starting Grant to M.W. (101077037) and by multiple grants from the German Federal Ministry of Education and Research to B.K. (CLINSPECT-M, 161L0214A and 16LW0243K; ProteomeTools, 031L0008A) and M.F. (ESTHER, 13GW0603B). The funders had no role in study design, data collection and analysis, decision to publish or preparation of the manuscript.

### Author contributions
M.F. and M.W. conceived the study. M.F., M.W., and D.P.Z. developed and evaluated the initial prototype. F.S., P.S., T.S., M.G., I.B., S.B.F. and S.G. developed, implemented and optimized the algorithms. S.G., V.S., S.B.F., L.M. and M.G. developed the deep-learning models. T.S., M.S. and F.S. orchestrated the implementation of software modules and the deployment of the software. M.F., D.P.Z., F.S., M.G., M.T.B. and A.H. evaluated the algorithm. M.F., M.T.B., J.T., A.H. and D.P.Z. processed the results data. M.F., M.T.B., J.T., A.H. and D.P.Z. performed the data analysis. L.E. helped in the preparation of the figures. M.F., D.P.Z., M.T.B., A.H., F.S., P.S., S.G., T.S., J.T., B.K. and M.W. provided critical feedback, discussed the results and consulted in revisions. M.F., D.P.Z., J.T., B.K. and M.W. wrote the manuscript.

### Funding

## Competing interests

M.F., D.P.Z., S.G. and T.S. are co-founders, shareholders and employees of MSAID, a company that develops software for proteomics, including the algorithm presented in this manuscript. M.W. and B.K. are co-founders and shareholders of MSAID and OmicScouts, which operates in the field of proteomics, but they have no operational role in either company. M.T.B., A.H., F.S., M.G., P.S., S.B.F., V.S., L.E., I.B., L.M. and M.S. are employees of MSAID. MSAID is an applicant on multiple pending patent applications covering functionality implemented in CHIMERYS that list M.W., M.F., T.S. and F.S. as inventors. The remaining authors declare no competing interests.

## Additional information

**Extended data** is available for this paper at

**Supplementary information** The online version
contains supplementary material available at

**Correspondence and requests for materials** should be addressed to
Martin Frejno or Mathias Wilhelm.

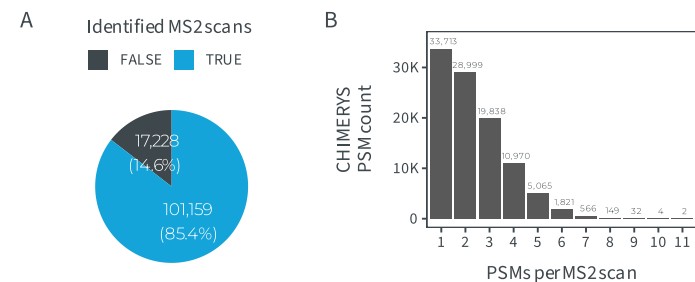

**Extended Data Fig. 1 | Chimeric DDA spectra. (A)** Proportions of MS2 spectra with at least one (blue) or no PSM identification (gray) in a 2-h HeLa DDA single-shot measurement (n = 1). Data taken from the LFQbench-type dataset and acquired on an Orbitrap QE HF-X with 1.3 Th isolation windows. **(B)** Distribution of the number of PSMs per MS2 spectrum for the same data as in (**A**).

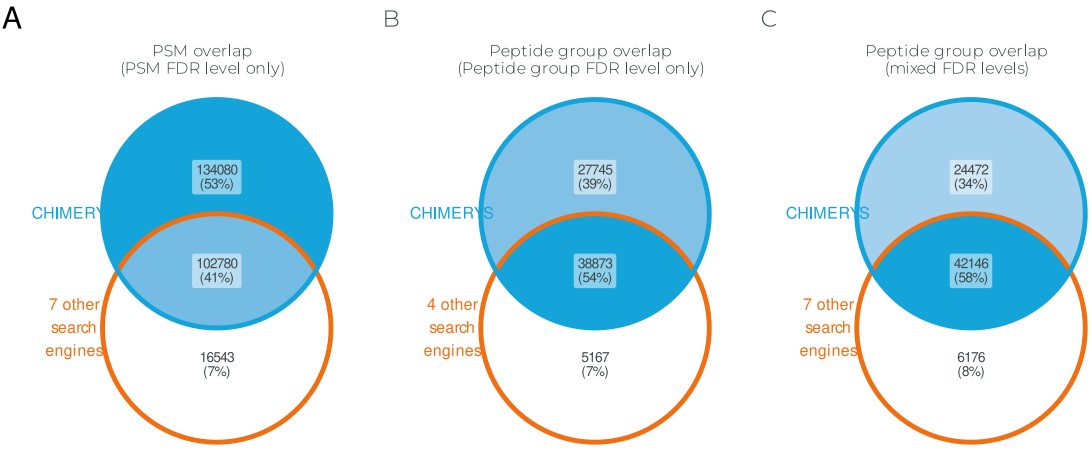

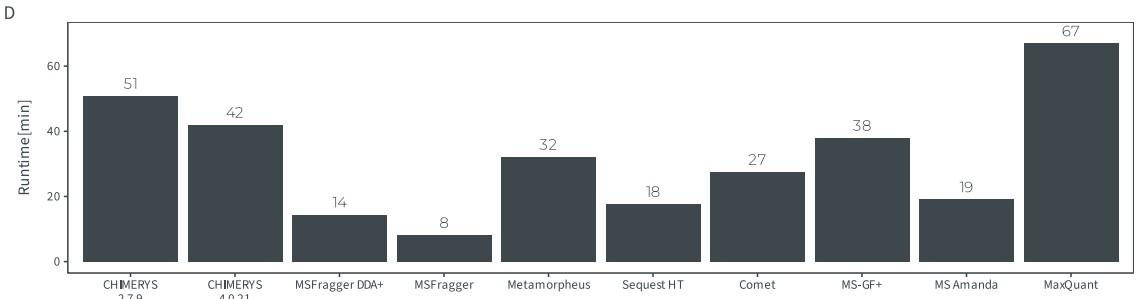

**Extended Data Fig. 2 | Overlap in identifications and runtime comparison.** Venn diagram of PSM (**A**) and peptide group (**B-C**) identifications in a 2-h HeLa DDA single-shot measurement (n = 1) comparing CHIMERYS to the combination of other search engines at 1% PSM-level FDR (**A**), 1% peptide group-level FDR (**B**) or 1% FDR at different levels (**C**). The different FDR levels in (**C**) were the peptide group level for CHIMERYS, Sequest HT, Comet, MS Amanda and MaxQuant, the precursor level for MSFragger and the PSM level for Metamorpheus and MS-GF + . Data was acquired on an Orbitrap QE HF-X with 1.3 Th isolation windows and taken from the LFQbench-type dataset. (**D**) Runtime comparison for the analyses shown in (**A**). All search engines were run in a virtual Windows 10 environment with 8 cores and 64 GB of RAM to mirror the cloud environment of CHIMERYS.

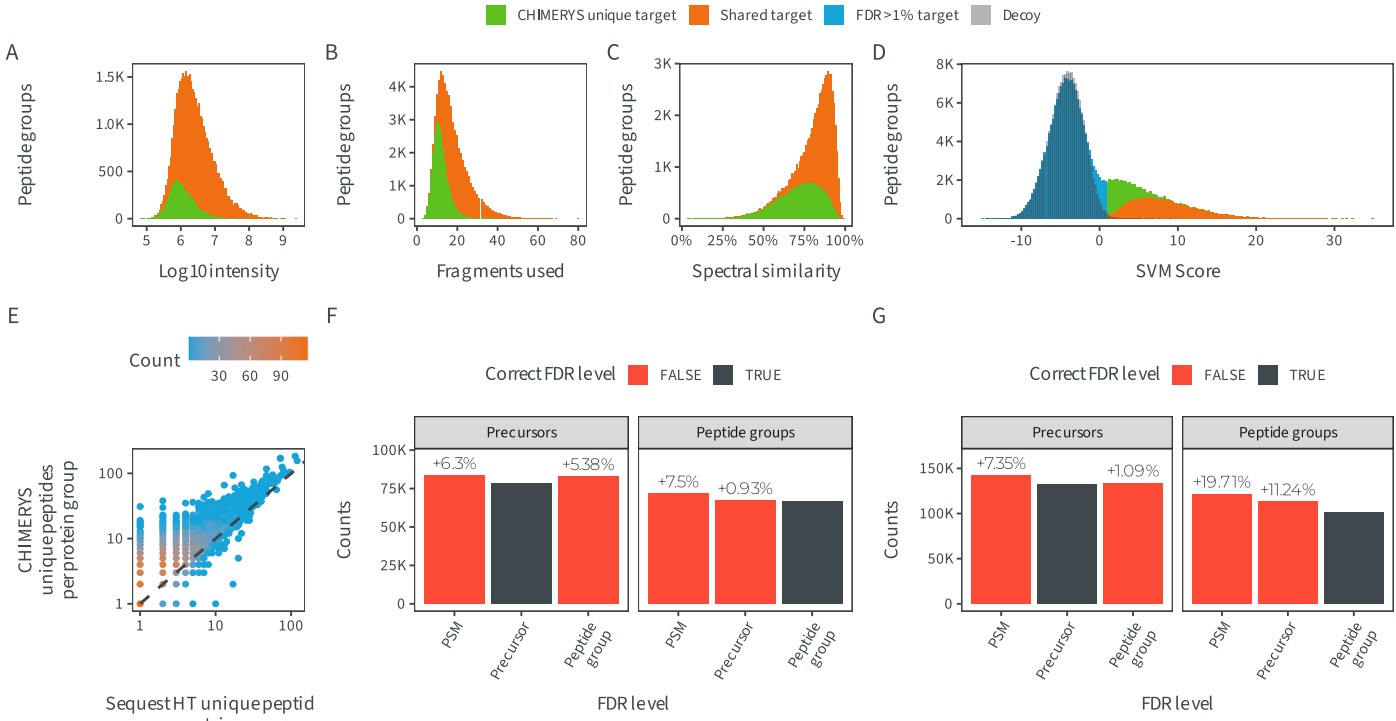

**Extended Data Fig. 3 | CHIMERYS-unique identifications and FDR levels.**
**(A)** MS1 apex intensity distribution of peptide groups filtered at 1% global peptide group-level FDR and identified uniquely by CHIMERYS (green) or also by at least one other search engine tested as part of Fig. 1e (orange). The dataset was a 2-h HeLa DDA single-shot measurement (n = 1) from the LFQbench-type dataset, acquired on an Orbitrap QE HF-X with 1.3 Th isolation windows. **(B)** Distribution of the number of matched peaks between predicted and experimental spectra for the same data as in (**A**). **(C)** Distribution of the normalized spectral contrast angle after deconvolution between predicted and experimental spectra for the same data as in (**A**). **(D)** Distribution of the support vector machine score from mokapot for the same data as in (**A**), including targets with a global peptide group-level FDR exceeding 1% (blue). The support vector machine score

distribution for decoys (gray) is overlayed. **(E)** Scatter-plot of the number of unique peptides per protein group identified by Sequest HT (x axis) or CHIMERYS (y axis) for the same data as in (**A**). Protein groups are filtered to 1% global protein FDR and peptides are filtered to 1% global peptide group-level FDR. **(F)** The number of precursors (left) or peptide groups (right) identified by CHIMERYS at 1% run-specific PSM-, run-specific precursor- or global peptide group-level FDR. The dataset is a single 2-h HeLa DDA single-shot measurement (n = 1), acquired on an Orbitrap QE HF-X with 1.3 Th isolation windows and taken from the LFQbench-type dataset. **(G)** Same as (**F**), but for 2-h DDA single-shot measurements from two different conditions, acquired in three replicates on an Orbitrap QE HF-X with 1.3 Th isolation windows from the LFQbench-type dataset (n = 6).

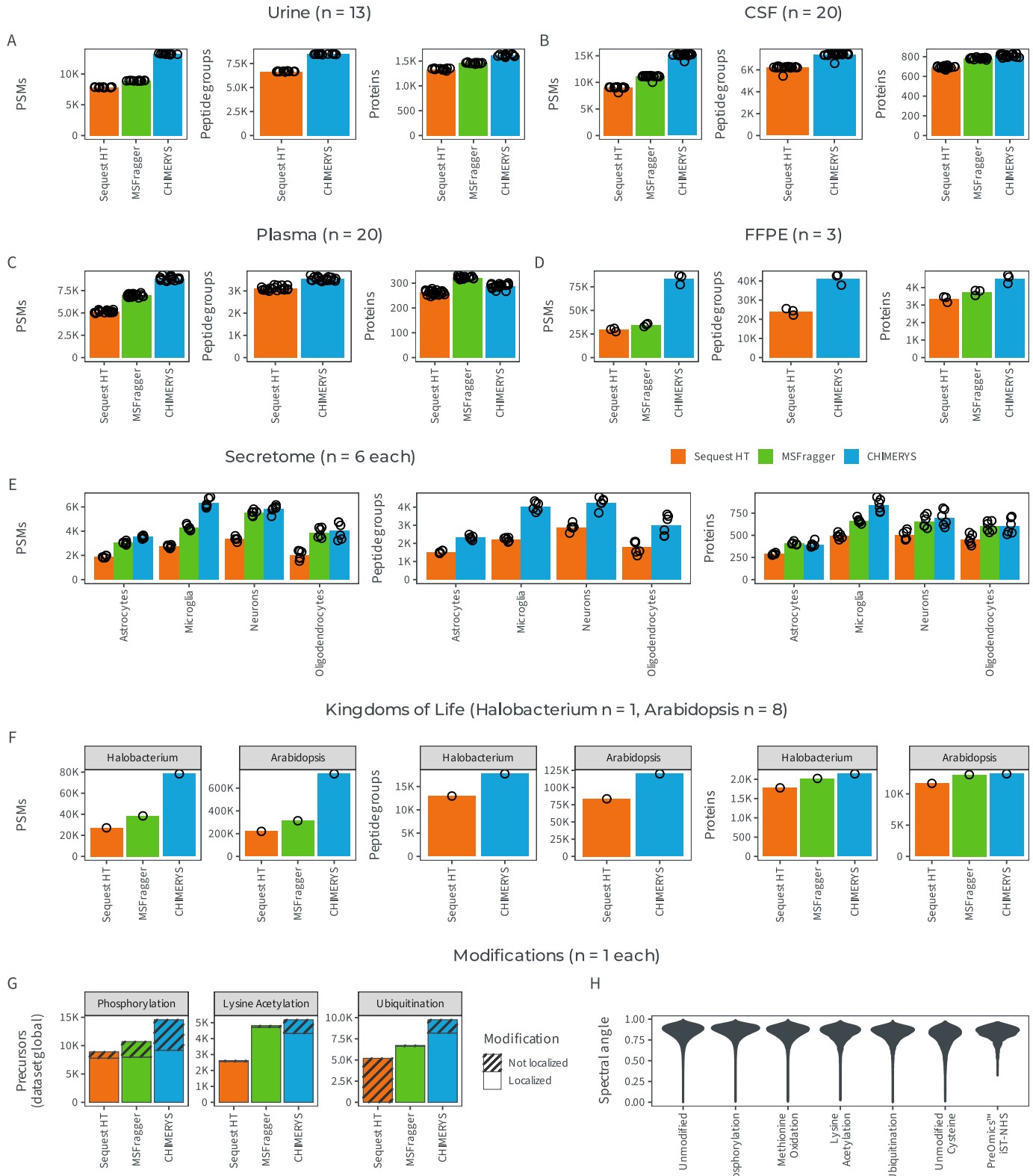

**Extended Data Fig. 4 | Comparison of CHIMERYS to Sequest HT and MSFragger on various datasets.** PSM, peptide and protein group identifications based on Sequest HT (orange), MSFragger (green) and CHIMERYS (blue) from measurements of human urine[32] **(A)**, CSF[32] **(B)** and plasma[32] **(C)**, FFPE **(D)** and secretome samples[33] **(E)**, as well as from publicly available 1-h measurements of *Arabidopsis thaliana* and *Halobacterium*[34] **(F)**. FDR was controlled at the run-specific PSM, peptide group (only available for Sequest HT and CHIMERYS) and protein group level, respectively. **(G)** Comparison of phosphorylated[35], acetylated[36] and ubiquitinated[36] precursors for Sequest HT (orange), MSFragger (green) and CHIMERYS v.4.0.21 (blue). FDR was controlled at the run-specific precursor level. The shaded proportion of the barchart displays precursors with a localization probability of >0.7 as calculated by ptmRS for Sequest HT, MSFragger or CHIMERYS (native localization). Note that ptmRS does not support ubiquitination. **(H)** Violin plot of the spectral angle comparing fragment ion predictions of INFERYS v.4.0.0 for unmodified and modified peptides to a hold-out dataset.

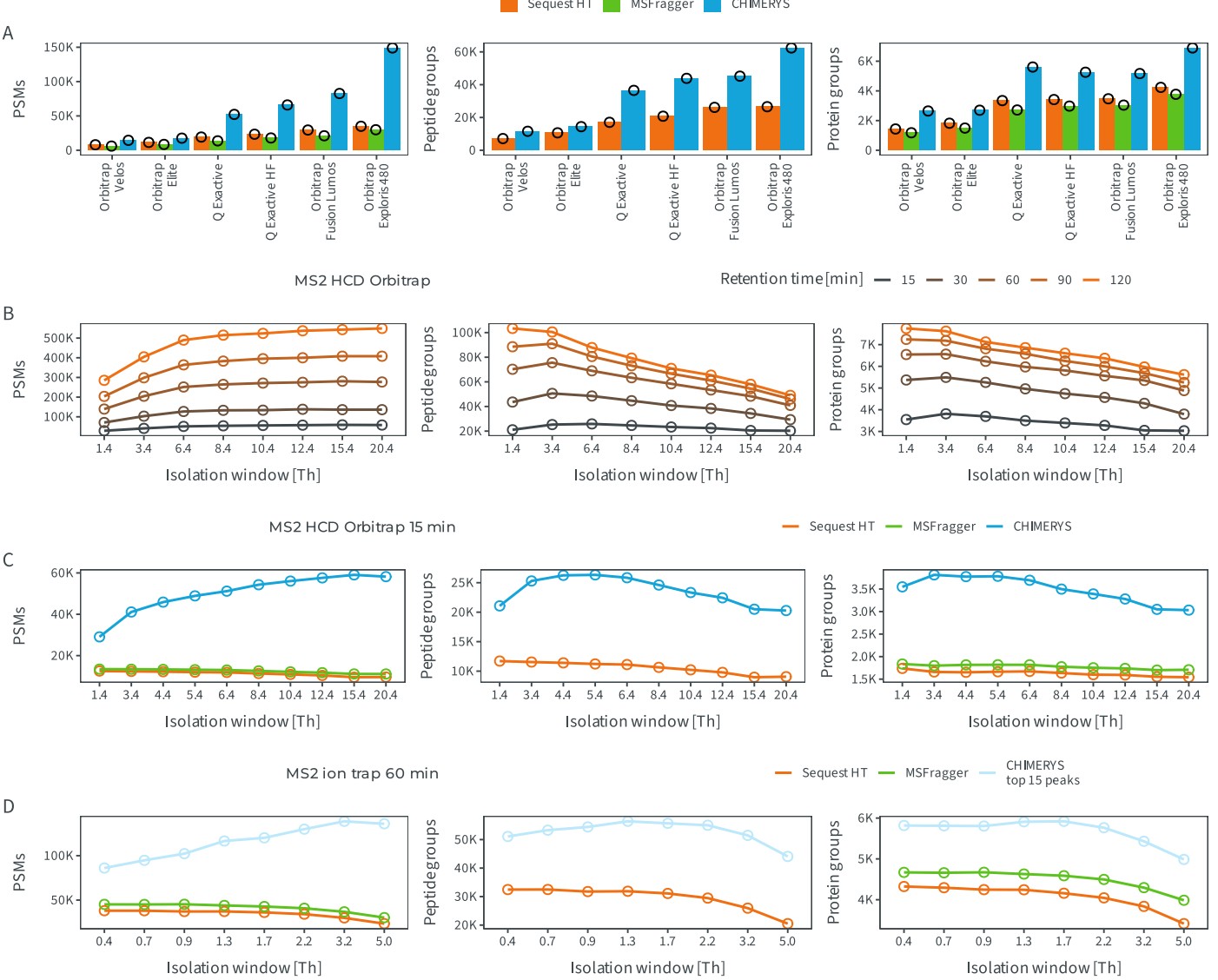

**Extended Data Fig. 5 | Instrument generations, gradients and wwDDA/WWA. (A)** PSM, peptide and protein group identifications based on Sequest HT (orange), MSFragger (green) and CHIMERYS (blue) from 1-h HeLa single-shot measurements, acquired using various Orbitrap generations (n = 1). FDR was controlled at 1% at the run-specific PSM-, global peptide group- (only available for Sequest HT and CHIMERYS) and global protein level, respectively. **(B)** PSM, peptide and protein group identifications based on CHIMERYS from pancreatic mouse cell single-shot measurements, acquired using different gradient lengths and isolation window widths (n = 1). FDR was controlled at 1% at the run-specific PSM-, global peptide group- and global protein level, respectively. **(C)** PSM, peptide and protein group identifications based on Sequest HT (orange), MSFragger (green) and CHIMERYS (blue) from 15 min pancreatic mouse cell single-shot measurements (n = 1). Data was acquired using HCD fragmentation with Orbitrap readout and different isolation window widths. FDR was controlled at 1% at the run-specific PSM-, global peptide group- (only available for Sequest HT and CHIMERYS) and global protein level, respectively. **(D)** PSM, peptide and protein group identifications based on Sequest HT (orange), MSFragger (green) and CHIMERYS after removal of low-abundance peaks (light blue) from 1-h HeLa single-shot measurements (n = 1). Data was acquired using CID fragmentation with ion trap readout and different isolation window widths. FDR was controlled at 1% at the run-specific PSM-, global peptide group- (only available for Sequest HT and CHIMERYS) and global protein level, respectively.

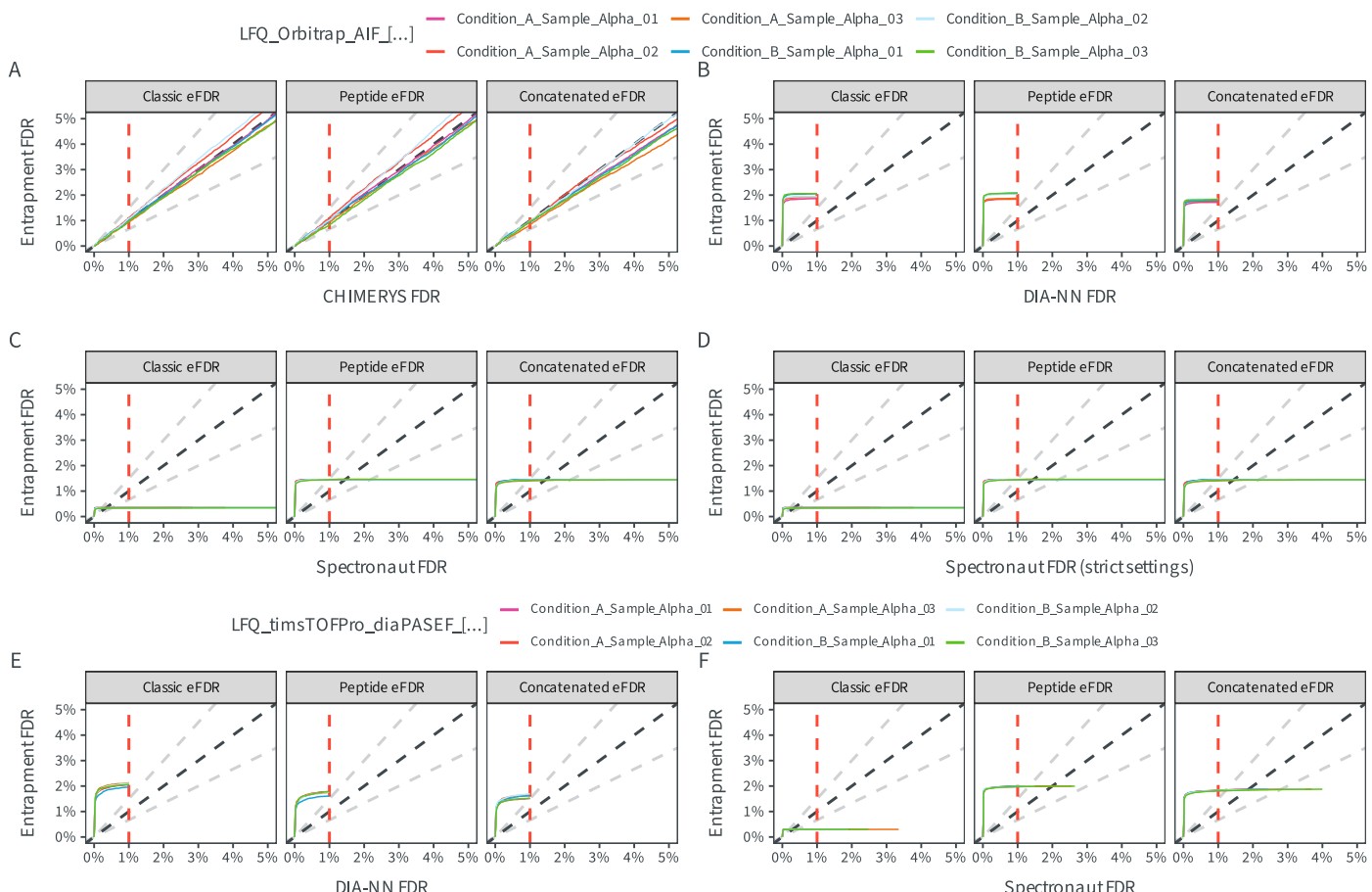

**Extended Data Fig. 6 | Entrapment analyses on DIA data.** Scatter plots of run-specific precursor-level self-reported (x axis) and entrapment FDR (y axis) from three different entrapment approaches (Supplementary Methods). Data is shown for CHIMERYS **(A)**, DIA-NN **(B)**, Spectronaut with default settings **(C)** and Spectronaut with more stringent settings[41] **(D)** on triplicate 2-h DIA single-shot measurements from two different conditions (n = 6). Data was acquired on an Orbitrap QE HF-X with 8 Th isolation windows and taken from the LFQbench-type dataset[37]. **(E)** same as in **(B)**, but for the corresponding TimsTOF Pro data. **(F)** same as in **(C)**, but for the corresponding TimsTOF Pro data.

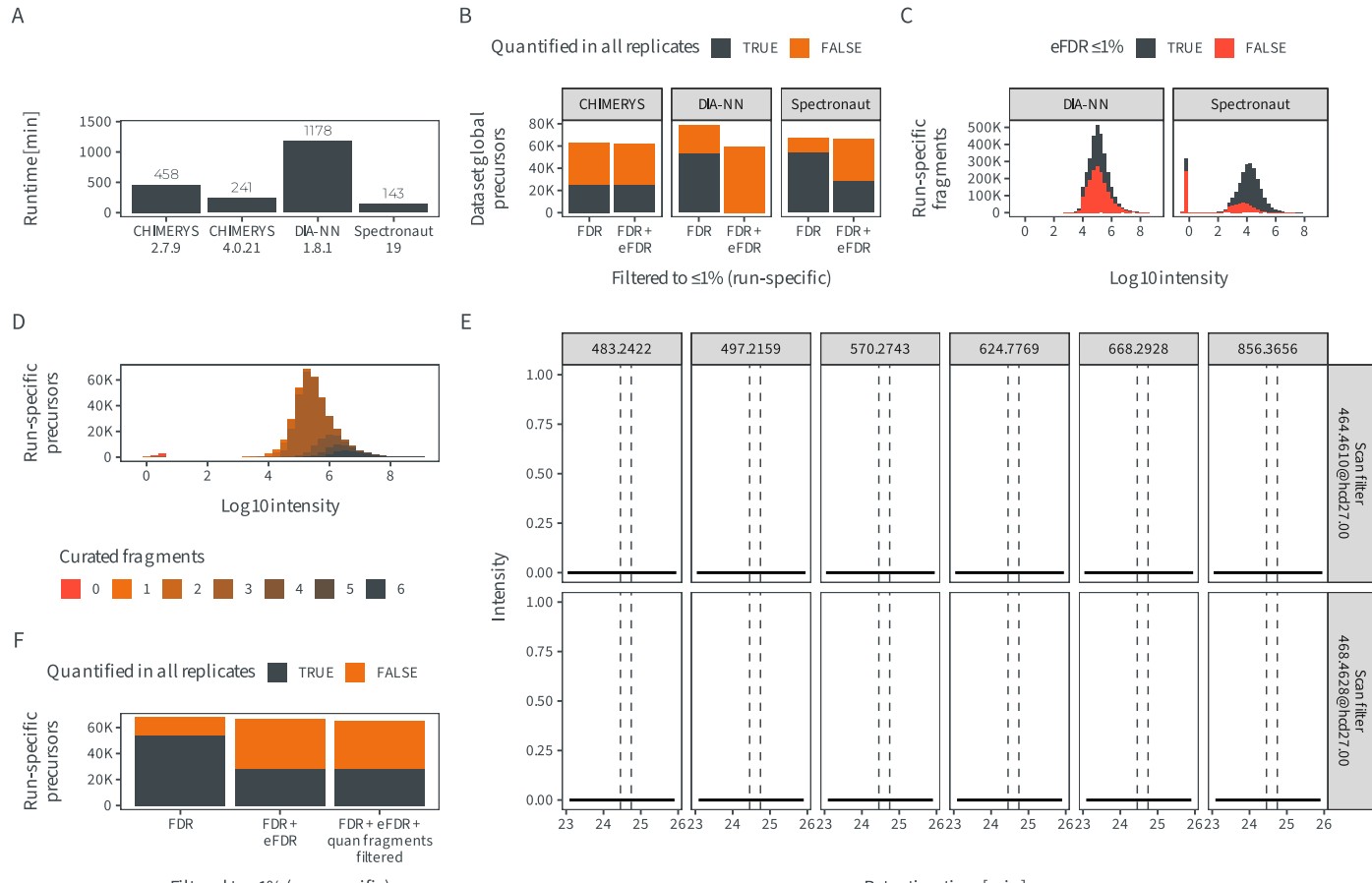

**Extended Data Fig. 7 | DIA data analysis with CHIMERYS, DIA-NN and Spectronaut. (A)** Runtime comparison of CHIMERYS v.2.7.9, CHIMERYS v.4.0.21, DIA-NN v.1.8.1 and Spectronaut v.19 for the peptide eFDR approach. Runtimes include spectral library generation. **(B)** Precursors quantified by CHIMERYS, DIA-NN and Spectronaut in at least one (orange) or three (gray) out of three replicate measurements of two different conditions (n = 6) in a multispecies LFQbench dataset. Identifications are filtered at 1% run-specific precursor-level FDR or additionally at 1% run-specific precursor-level eFDR based on the peptide eFDR approach (Supplementary Methods). **(C)** Peak areas from DIA-NN and Spectronaut for fragment ions from precursors surviving (gray) or not surviving (red) 1% run-specific precursor-level eFDR based on the peptide eFDR approach for the same data as in **(B)** **(D)** Apex intensities for precursors identified by Spectronaut at 1% run-specific precursor-level FDR for the same data as in **(B)**. Precursors are colored by the number of fragment ions with Peak

areas (F.PeakArea) > 1 that were not excluded from quantification by Spectronaut (curated fragments). **(E)** Example fragment ion XICs directly extracted from the raw file for the precursor at m/z 466.9506 identified by Spectronaut but not by CHIMERYS in Fig. 3d. All six library fragments are shown. XICs were extracted using the R package rawrr with 20 ppm fragment mass tolerance. **(F)** Precursors quantified by Spectronaut in at least one (orange) or three (gray) out of three replicate measurements of two different conditions in a multispecies LFQbench dataset (n = 6). Identifications in the 1st bar are filtered at 1% run-specific precursor-level FDR. Additionally, the 2nd bar is filtered at 1% run-specific precursor-level eFDR based on the peptide eFDR approach. Additionally, the 3rd bar is filtered by excluding precursors that are quantified based on less than three fragment ions with peak areas (F.PeakArea) > 1, which were not excluded from quantification by Spectronaut.

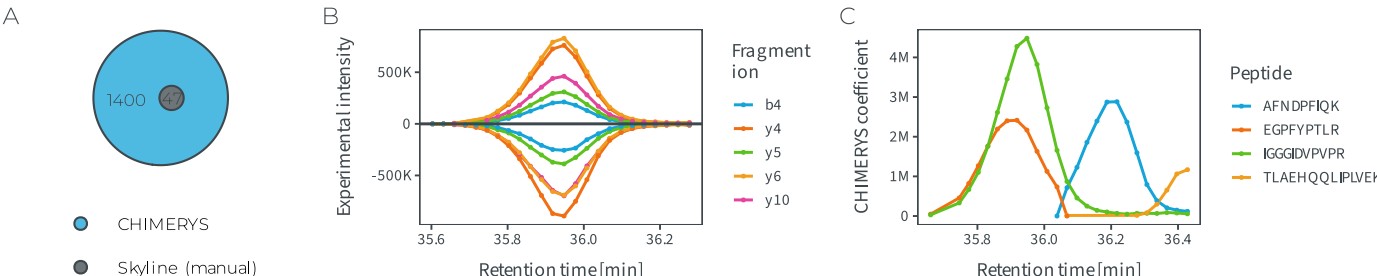

**Extended Data Fig. 8 | PRM and DIA quantification using CHIMERYS coefficients.** (A) Venn diagram of peptides identified in a PRM dataset (n = 1) – targeting 52 peptides from 18 human proteins – by CHIMERYS at 1% peptide group-level FDR (blue) or Skyline (gray). (B) Mirror XIC of the top five experimental (above the x axis) and predicted fragment ion intensities, scaled by the corresponding CHIMERYS coefficients (below the x axis) for one of the targeted peptides in **A**. (C) Coefficient-based reconstruction of elution peaks for four different peptides identified by CHIMERYS in the data in (**A**), only one of which was targeted in the assay (IGGGIDVPVPR).

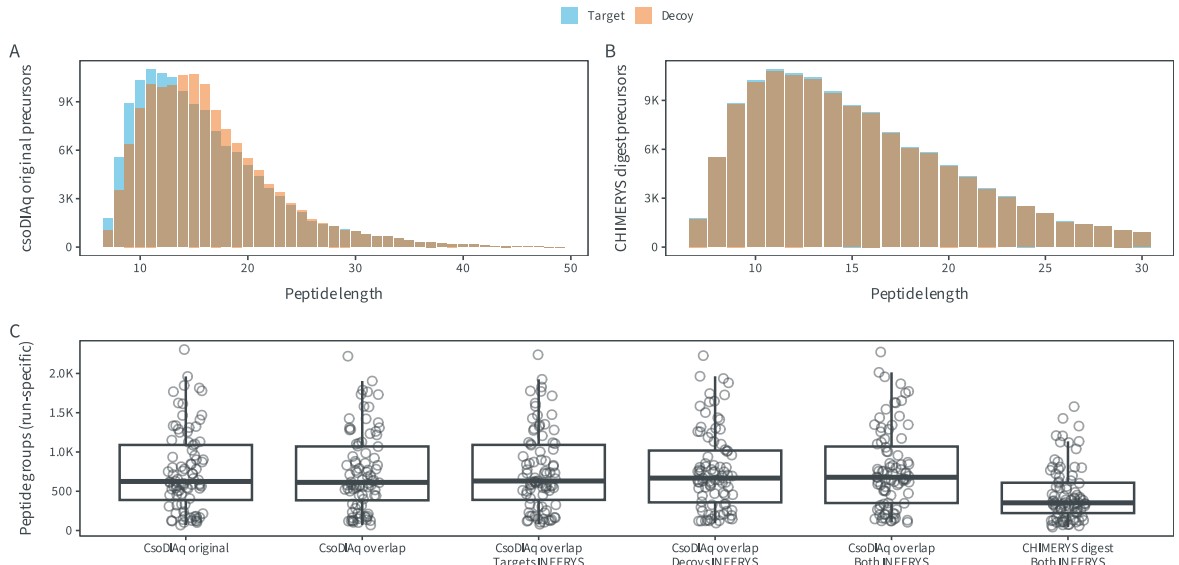

**Extended Data Fig. 9 | CsoDIAq results as a function of the chosen library.**
**(A-B)** Peptide length distributions for **(A)** the original DI-SPA library and **(B)** a library generated using the original DI-SPA library targets and decoys generated by CHIMERYS. **(C)** Boxplot visualizing the number of peptide groups identified by CsoDIAq at 1% run-specific FDR in each sample using different libraries (individual boxplots), including the ones shown in **(A)** and **(B)** (first and last, respectively, see also Supplementary Methods). The bottom (lower hinge), center and top of the box (upper hinge) are the first, second (median) and third quartile of the data points. The upper whisker extends from the hinge to the largest value no further than 1.5 * IQR from the hinge (IQR is the inter-quartile range, that is the distance between the first and third quartiles). The lower whisker extends from the hinge to the smallest value at most 1.5 * IQR of the hinge. All individual data points are shown (n = 88 raw files).

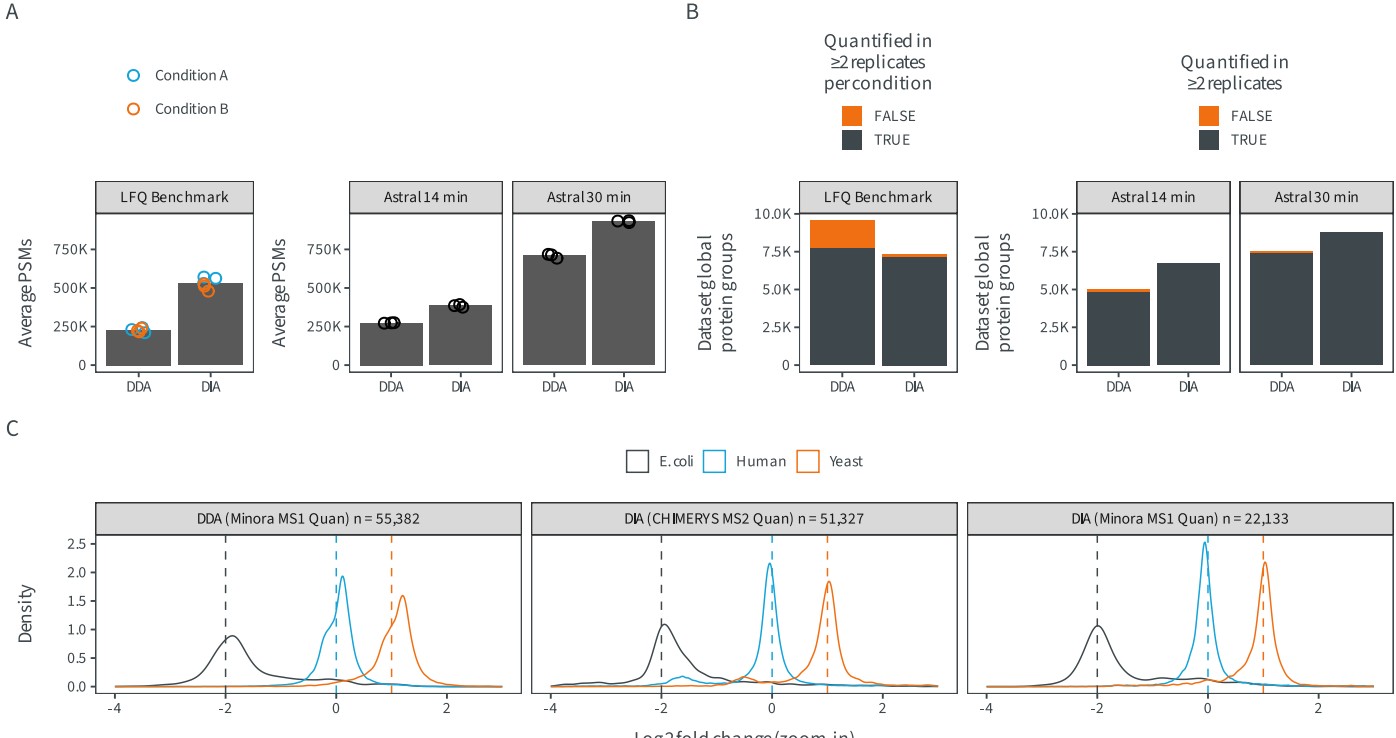

**Extended Data Fig. 10 | Comparison of DDA and DIA data.** PSMs **(A)** and protein groups **(B)** identified by CHIMERYS in DDA or DIA data. For the LFQbench-type dataset, triplicate 2-h single-shot measurements from two conditions (n = 6) are shown. Data was acquired on an Orbitrap QE HF-X with 1.3 or 8 Th isolation windows for DDA and DIA, respectively. For the Orbitrap Astral datasets, triplicate 14 min or 30 min single-shot measurements from a HeLa sample (n = 3) are shown. FDR was controlled at 1% at the run-specific PSM level or at the global protein level, respectively. Match between runs was used for DDA data and for DIA data, peptide groups were quantified irrespective of their run-specific FDR. **(C)** Peptide group-level log$_2$-ratio density plots for the same DDA and DIA data from the LFQbench dataset as in **(A)**, quantified in MS1 using the Minora Feature Detector or in MS2 using CHIMERYS.

# Reporting Summary

## Statistics

For all statistical analyses, confirm that the following items are present in the figure legend, table legend, main text, or Methods section.

| n/a | Confirmed | |
|---|---|---|
| ☐ | ☒ | The exact sample size (*n*) for each experimental group/condition, given as a discrete number and unit of measurement |
| ☐ | ☒ | A statement on whether measurements were taken from distinct samples or whether the same sample was measured repeatedly |
| ☒ | ☐ | The statistical test(s) used AND whether they are one- or two-sided<br>*Only common tests should be described solely by name; describe more complex techniques in the Methods section.* |
| ☒ | ☐ | A description of all covariates tested |
| ☒ | ☐ | A description of any assumptions or corrections, such as tests of normality and adjustment for multiple comparisons |
| ☒ | ☐ | A full description of the statistical parameters including central tendency (e.g. means) or other basic estimates (e.g. regression coefficient) AND variation (e.g. standard deviation) or associated estimates of uncertainty (e.g. confidence intervals) |
| ☒ | ☐ | For null hypothesis testing, the test statistic (e.g. *F*, *t*, *r*) with confidence intervals, effect sizes, degrees of freedom and *P* value noted<br>*Give P values as exact values whenever suitable.* |
| ☒ | ☐ | For Bayesian analysis, information on the choice of priors and Markov chain Monte Carlo settings |
| ☒ | ☐ | For hierarchical and complex designs, identification of the appropriate level for tests and full reporting of outcomes |
| ☐ | ☒ | Estimates of effect sizes (e.g. Cohen's *d*, Pearson's *r*), indicating how they were calculated |

*Our web collection on statistics for biologists contains articles on many of the points above.*

## Software and code

Policy information about availability of computer code

Data collection | The specific software versions used for LC-MS/MS raw data acquisition is available as meta information in the corresponding raw data.

Data analysis | Raw mass spectrometry data was analyzed with CHIMERYS 2.7.9 from PD 3.1.0.622 or PD 3.1.0.638 using INFERYS 3.0.0, CHIMERYS 4.0.21 from a pre-release version of PD 3.2 using INFERYS 4.0.0, Sequest HT as available in PD 3.1.0.622 or PD 3.1.0.638, MS Amanda 3.1.21.532 from PD 3.1.0.638, Comet 2019.01 rev. 1 as available in PD 3.1.0.622 or 3.1.0.638, MSFragger 4.0 with Philosopher 5.1.0 from FragPipe 21.1 ("Default" and "WWA" workflow with "DDA+" data type), MetaMorpheus 1.0.5, MS-GF+ 2024.03.2, MaxQuant 2.4.2.0 or 2.6.5.0, DIA-NN 1.8.1, Spectronaut 19, Skyline 22.2 and CsoDIAq 2.1.2. Raw files were converted into mzXML files using MSConvert 3.0.23121-9c54301. In silico digest of fasta files was performed using Protein Digestion Simulator 2.4.7993.32903. Models were built and trained with Tensorflow 2.11.1. The Mokapot version used in this study is available on GitHub (https://github.com/wfondrie/mokapot/). The modifications to the mimic entrapment database generator are available on GitHub (https://github.com/percolator/mimic/). A web-version of the mimic tool can be found at https://mimic.msaid.io/. A demo version of Proteome Discoverer and CHIMERYS can be requested at https://www.msaid.de/chimerys-demo or by contacting the corresponding authors. The custom R scripts used for data analysis are available on GitHub (https://github.com/msaid-de/chimerys-manuscript). KEGG pathway enrichment was performed using Cytoscape 3.10.3 with the STRING plugin 2.1.1.

For manuscripts utilizing custom algorithms or software that are central to the research but not yet described in published literature, software must be made available to editors and reviewers. We strongly encourage code deposition in a community repository (e.g. GitHub). See the Nature Portfolio guidelines for submitting code & software for further information.

## Data

Policy information about availability of data

All manuscripts must include a data availability statement. This statement should provide the following information, where applicable:
- Accession codes, unique identifiers, or web links for publicly available datasets
- A description of any restrictions on data availability
- For clinical datasets or third party data, please ensure that the statement adheres to our policy

Data Availability

External raw data

The following external data were downloaded from PRIDE or MassIVE and processed with the respective search engines. In brief, body fluid data from Bian et al., 2020 (PXD015087), secretome data from Tüshaus et al., 2020 (PXD018171), Arabidopsis and Halobacterium data from Müller et al., 2020 (PXD014877), phosphorylation data from Frejno et al., 2020 (PXD013615), acetylation and ubiquitination data from Zecha et al., 2022 (PXD023218), triple species mix as well as HeLa data from the LFQBench-type dataset by Van Puyvelde et al., 2022 (PXD028735), Orbitrap Astral data extracted from Gutzman et al., 2024 (PXD046453) and DI-SPA data from Meyer et al. (MSV000085156). Notably, peptides containing methionine residues were excluded from all analyses of the LFQBench-type dataset, since raw files might show differential oxidation. The same applies to the Orbitrap Astral data from PXD046453. For the LFQBench-type dataset, technical replicates were analyzed (see Supplementary Table 3). All other replicates are biological replicates. An itemized mapping of external data processed as part of this study to their source is available in Supplementary Table 3.

Internal raw data

The following datasets were generated in house: FFPE (biological replicates), gradient comparison, wwDDA, instrument generations and PRM data. An overview of the files generated is provided in Supplementary Table 4. The generated mass spectrometric raw and search data of internal datasets from this study are available via PRIDE with the dataset identifier PXD053241.

Fasta files

All fasta files used in this study are available via PRIDE with the dataset identifier PXD053241.

Source Data

All Source and Supplementary Data files required to reproduce this study are available via PRIDE with the dataset identifier PXD053241.

## Human research participants

Policy information about studies involving human research participants and Sex and Gender in Research.

| | |
|---|---|
| Reporting on sex and gender | n/a |
| Population characteristics | n/a |
| Recruitment | n/a |
| Ethics oversight | n/a |

Note that full information on the approval of the study protocol must also be provided in the manuscript.

# Field-specific reporting

Please select the one below that is the best fit for your research. If you are not sure, read the appropriate sections before making your selection.

☒ Life sciences          ☐ Behavioural & social sciences          ☐ Ecological, evolutionary & environmental sciences

For a reference copy of the document with all sections, see nature.com/documents/nr-reporting-summary-flat.pdf

# Life sciences study design

All studies must disclose on these points even when the disclosure is negative.

| | |
|---|---|
| Sample size | No sample size calculation was performed. The main focus of this study is to introduce CHIMERYS and compare it to other software packages, which could be done on individual samples. |
| Data exclusions | Peptides containing methionine residues were excluded from all analyses of the LFQBench-type dataset, since raw files might show differential oxidation. The same applies to the Orbitrap Astral data from PXD046453. |

| | |
|---|---|
| Replication | From publicly available datasets, three replicates were analyzed to assess data variability. The number of replicates is indicated in the corresponding figure legends. All replication attempts for data generated in house were successful. |
| Randomization | Randomization was not applicable to this study since no biological conditions were compared (e.g. treatment versus control). |
| Blinding | Blinding was not applicable to this study since no biological conditions were compared (e.g. treatment versus control). |

# Reporting for specific materials, systems and methods

We require information from authors about some types of materials, experimental systems and methods used in many studies. Here, indicate whether each material, system or method listed is relevant to your study. If you are not sure if a list item applies to your research, read the appropriate section before selecting a response.

## Materials & experimental systems

| n/a | Involved in the study |
|---|---|
| ☒ | ☐ Antibodies |
| ☐ | ☒ Eukaryotic cell lines |
| ☒ | ☐ Palaeontology and archaeology |
| ☒ | ☐ Animals and other organisms |
| ☒ | ☐ Clinical data |
| ☒ | ☐ Dual use research of concern |

## Methods

| n/a | Involved in the study |
|---|---|
| ☒ | ☐ ChIP-seq |
| ☒ | ☐ Flow cytometry |
| ☒ | ☐ MRI-based neuroimaging |

## Eukaryotic cell lines

Policy information about cell lines and Sex and Gender in Research

| | |
|---|---|
| Cell line source(s) | HeLa cells were sourced from ATCC (CCL-2). |
| Authentication | None of the cell lines used were authenticated. No biological conclusions were drawn based on the data. |
| Mycoplasma contamination | Cell lines were not tested for mycoplasma contamination. No biological conclusions were drawn based on the data. |
| Commonly misidentified lines (See ICLAC register) | The DI-SPA data from Meyer et al. (MSV000085156) used MCF7 cells. No biological conclusions were drawn based on the data. |

