## [Peer Review File · Nature Methods]

Unifying the analysis of bottom-up proteomics data with CHIMERYS

Corresponding Author: Dr Martin Frejno

Version 0:

Decision Letter:

21st Aug 2024

Dear Dr. Frejno,

Your Article, "Unifying the analysis of bottom-up proteomics data with CHIMERYS", has now been seen by 2 reviewers. As you will see from their comments below, although the reviewers find your work of considerable potential interest, they have raised a number of concerns. We are interested in the possibility of publishing your paper in Nature Methods, but would like to consider your response to these concerns before we reach a final decision on publication.

We therefore invite you to revise your manuscript to address these concerns. It would be important to demonstrate that the approach helps gain biological insights not otherwise obtainable when using existing methods. We would also like all technical concerns to be satisfactorily addressed.

Link Redacted

We hope to receive your revised paper within 6 weeks. If you cannot send it within this time, please let us know. In this event, we will still be happy to reconsider your paper at a later date so long as nothing similar has been accepted for publication at Nature Methods or published elsewhere.

OPEN SCIENCE REQUIREMENTS

REPORTING SUMMARY AND EDITORIAL POLICY CHECKLISTS

IMAGE INTEGRITY

DATA AVAILABILITY

All novel DNA and RNA sequencing data, protein sequences, genetic polymorphisms, linked genotype and phenotype data, gene expression data, macromolecular structures, and proteomics data must be deposited in a publicly accessible database, and accession codes and associated hyperlinks must be provided in the "Data Availability" section.

CODE AVAILABILITY

Please include a "Code Availability" subsection in the Online Methods which details how your custom code is made available. Only in rare cases (where code is not central to the main conclusions of the paper) is the statement "available upon request" allowed (and reasons should be specified).

MATERIALS AVAILABILITY

SUPPLEMENTARY PROTOCOL

To help facilitate reproducibility and uptake of your method, we ask you to prepare a step-by-step Supplementary Protocol for the method described in this paper. We [encourage authors to share their step-by-step experimental protocols](https://www.nature.com/nature-research/editorial-policies/reporting-standards#protocols) on a protocol sharing platform of their choice and report the protocol DOI in the reference list. Nature Portfolio's protocols.io is a free-to-use and open resource for protocols; protocols deposited onto protocols.io are citable and can be linked from the published article. More details can found at [protocols.io](https://www.protocols.io/help/publish-articles).

ORCID

Sincerely,
Arunima

Arunima Singh, Ph.D.
Senior Editor
Nature Methods

Reviewers' Comments:

Reviewer #1 (Remarks to the Author):

In this manuscript, the authors present a bottom-up proteomics data analysis software, CHIMERY5. It applies a spectrum-centric approach to unify peptide identification and quantification for data-dependent acquisition (DDA), data-independent acquisition (DIA), and parallel reaction monitoring (PRM). In contrast to most traditional spectrum-centric approaches, CHIMERY5 heavily incorporates the deep-learning prediction of fragment intensities and retention time into the database searching and quantification procedures, which largely boosts the sensitivity, but also introduces limitations. To demonstrate the performance of CHIMERY5, the authors used the existing DDA, DIA, and PRM data analysis tools as benchmarks. More importantly, the authors applied the entrapment database search to extensively evaluate the false discovery rate (FDR) of peptide identification performed by CHIMERY5 and other DIA tools. To benchmark the quantification results, they applied the LFQbench-type dataset to evaluate precision and accuracy. This manuscript has been written and organized. The proposed methods and CHIMERY5 software are promising. I am glad to see that people begin to care about quality over quantity in identification and quantification. However, several major and minor issues must be addressed.

Major issues

1. Unifying the analysis of DDA, DIA, and PRM is one of the major contributions of this work. From my point of view, this is not a significant scientific innovation. There have been tools, such as MSFragger and MSFragger-DIA, and MaxQuant and MaxLFQ, trying to unify the DDA and DIA data analysis. Furthermore, as far as I can see, there is no harm to analyzing DDA and DIA using different tools because people normally do not directly compare DDA with DIA in biological studies. It would be

better if the authors demonstrate some case studies that only the unified DDA and DIA analysis can do, while other approaches, such as using MSFragger for DDA and DIA-NN for DIA, will not be able to achieve.

2. Although CHIMERYs uses the spectrum-centric approach, it has the limitations and disadvantages specific to the peptide-centric approach owing to its reliance on prediction. The limitations include limited modifications, peptide length ranges, and number of missed cleavages. These disadvantages are mostly related to the long runtime and additional requirements of graphics processing unit (GPU) hardware. The authors have discussed those limitations in the discussion section: "A current limitation of CHIMERYs in this context is that peptides carrying modifications that are not yet covered by the underlying deep learning model escape detection. It can be anticipated that this limitation will diminish over time as deep learning models start to emerge that are capable of generalizing to modifications or fragmentation methods that they have not yet been trained for.". Unfortunately, there appears to be no ideal solution in the near future. Rescoring tools, such as MSBooster, overcome the modification limitation by stripping unsupported modifications during the prediction, and shifting the fragment peaks based on the modification masses after the prediction. Of course, this solution is not ideal because some modifications change the fragmentation and retention time. The authors should discuss more about the limitations and possible solutions if the desired deep learning models will not be available soon.

3. The authors did not compare runtimes among the CHIMERYs and other tools. This is an important benchmark because 1) the data scale is becoming increasingly larger with the development of mass spectrometry and sample preparation techniques, and 2) CHIMERYs is computationally intensive because it performs deep-learning predictions. The authors should perform the runtime comparison using the tools mentioned in Figure 1 and 3.

4. The current version of CHIMERYs only supports mass spectrometry data generated by the Thermo Fisher mass spectrometer. Do the authors have any plans to support the data from Bruker (especially timsTOF series), SCIEX, and Waters mass spectrometers?

5. In the Introduction section, the authors reviewed the peptide-centric approach for DIA data analysis, but missed the spectrum-centric approaches include DIA-Umpire (<https://doi.org/10.1038/nmeth.3255>), PECAN (<https://doi.org/10.1038/nmeth.4390>), MSFragger-DIA (<https://doi.org/10.1038/s41467-023-39869-5>), and diaTracer (<https://doi.org/10.1101/2024.05.25.595875>). It would be better to review them because the spectrum-centric approach is also what CHIMERYs is using.

6. The authors need to describe how they calculated the "CHIMERYs coefficient" because I could not find a clear description in either the Results or Online Methods section. In the Results section, they wrote "Apart from peptide identification, it also derives spectrum-centric quantitative information in the form of CHIMERYs coefficients, which can be interpreted as the interference-corrected total ion current for a given peptide in this MS2 spectrum (Online Methods). If none of the matched fragments for a peptide are shared with another peptide and the predicted MS2 spectrum matches perfectly to the experimental one, the coefficient is the sum of all matched fragment ions in the experimental MS2 spectrum. Hence, tracing the coefficient along retention time generates a pseudo-extracted-ion-chromatogram (XICs) that can be used to perform (relative) quantification of peptides based on their MS2 signal in PRM and DIA data.". What about there are matched fragments shared with another peptides? In addition, how can XIC be generated for DDA data with no fragment XICs?

7. In the Results section, the authors wrote "Briefly, predicted MS2 spectra from precursors with predicted retention times that fall within a data-dependent retention time window and precursor isotope envelopes that (partially) overlap with the isolation window are compared to experimental MS2 spectra.". Please elaborate more about the "data-dependent retention time window".

8. In Figure 1C, have the authors checked whether these peptides' predicted retention time is within a reasonable tolerance of the scan's observed retention time?

9. In Figure 3D, what are the criteria for selecting the precursors for the top and bottom panels? It might be unfair if pick a high-scored one identified by both Spectronaut and CHIMERYs while pick a low-scored one identified only by Spectronaut. Perhaps the authors should pick the lowest-scored one identified by both Spectronaut and CHIMERYs for the top panel and the highest-scored one identified only by Spectronaut for the bottom panel. It will support the authors' argument if the top one still has a higher quality than the lower one.

10. In the Results section, the authors wrote "To avoid differences in quantification due to different methods for determining peak integration borders, we compared the three algorithms based on their implementation of peak apex quantification.". I do not quite understand how they "compared the three algorithms based on their implementation of peak apex quantification". Did the authors re-implement the peak quantification algorithms of DIA-NN and Spectronaut? However, to the best of my knowledge, these algorithms are not publicly available.

11. In the Online Method section "data preprocessing-retention time prediction", the authors should clarify whether CHIMERYs 1) performs retention time alignment; and 2) uses apex retention time or the top-ranked PSM's precursor retention time.

Minor issues

1. The resolution of the figures seems to be lower than expected. It would be better if the authors double check them.

2. In the Online Methods section, the like of Thermo Fisher's RawFileReader (<http://planetorbitrap.com/rawfilereader>) does not seem correct. As far as I know, it should be <https://github.com/thermofisher/MSRawFileReader>.

3. In the Online Methods section, the authors wrote "Searches were performed using MSFragger v20.0 using the "Default" workflow. Additionally, MSFragger v21.1 was utilized with the "WWA" workflow and "DDA+" data type...". "v20.0" and "v21.1" are FragPipe's version numbers. The authors should also list the versions of MSFragger and Philosopher since they are used alone with FragPipe in the workflow.

4. In the "Code availability" section, the authors wrote "The custom R scripts used for data analysis are available upon request.". To increase the transparency of this scientific manuscript, it would be better to upload the scripts to GitHub.

Fengchao

Reviewer #2 (Remarks to the Author):

The authors of the manuscript have successfully shown the utility of the CHIMERYS algorithm for the identification and quantification of peptides from numerous bottom-up proteomics data types. The spectrum-centric approach, which integrates deep learning predictions with a parsimonious identification algorithm, represents a novel and promising advancement in the field. The data and methodology presented are comprehensive and robust, though there are a few areas that warrant revision.

Supplemental figure 4A-C: This figure is used to interpret results unique to CHIMERYS. However, the claim that CHIMERYS produces high-quality identifications while CHIMERYS specific results have a lower average spectral contrast angle requires further explanation. The authors should clarify why this metric is used as a supporting argument for their conclusion.

Optimization of Data Acquisition: The conclusions in the second paragraph of the section "Optimizing data acquisition with deconvolution in mind" would benefit from clarification. The discussion on gradually widening isolation windows, the resulting loss in peptide/protein sensitivity in CHIMERYS results, and the subsequent claim that CHIMERYS can counteract this effect, is somewhat unclear. It is not immediately evident which effect is being mitigated, especially since the described loss in sensitivity is a direct outcome of a CHIMERYS search.

Comparison with Other Software: Several questions arise in the comparison of CHIMERYS to other software. First, the rationale for selecting Sequest HT as the standard for comparison after an initial comparison with multiple software tools should be better explained. Additionally, some of the software versions used in the comparison appear to be several years out of date—the most up to date version of these software should be used to ensure a fair comparison.

CHIMERYS Coefficient: The CHIMERYS coefficient is described under ideal conditions with no shared fragment ions. The authors should address how this calculation is modified in the presence of shared fragment ions and how such conditions might impact quantification.

Server Setup for Predictions: The decision to set up a server for predictions, rather than allowing them to be performed locally, is not sufficiently justified. Providing a rationale for this choice would enhance the clarity of the methodology.

Clarity and Context:

- The use of the phrase "en passant" is unconventional and may be obscure to many readers. Given its primary definition relates to a chess move, the sentence in which it is used may be confusing. Consider replacing it with an English term or a more commonly understood phrase.
- The introduction discusses the subtractive approach to chimeric identifications and its limitations but does not address the multiplicative approach, where fragment ions are not removed for an additional search with a different precursor. This omission is further compounded in the results section, where the current handling of chimeric spectra is described as error-prone due to fragment ion peaks shared by multiple peptides.
- The statement, "A central challenge for data analysis lies in the fact that most MS2 spectra are chimeric, i.e., they contain more than one peptide because LC-MS/MS systems cannot fully separate the vast number of peptides resulting from whole proteome enzymatic digestion," is foundational to the paper but lacks citation and empirical support. The claim is supported only by Supplemental Figure 1, which is based on search results rather than the spectra themselves.
- The conclusions in the first paragraph of the "Comparison to other DIA search engines" section could be revised for greater clarity. The process by which the "Reduction to a comparable level" was achieved and the definition of "data completeness" are not fully explained.

Reviewer #2 (Remarks on code availability):

Percolator Mimic was readily available and easily understandable. Their version of mokapot requires a fair amount of github and coding knowledge to access.

Version 1:

Decision Letter:

Our ref: NMETH-A56765A

27th Nov 2024

Dear Martin,

Thank you for submitting your revised manuscript "Unifying the analysis of bottom-up proteomics data with CHIMERYS" (NMETH-A56765A). It has now been seen by the original referees and their comments are below. The reviewers find that the paper has improved in revision, and therefore we'll be happy in principle to publish it in Nature Methods, pending minor revisions to satisfy the referees' final requests and to comply with our editorial and formatting guidelines.

TRANSPARENT PEER REVIEW

ORCID

Sincerely,
Arunima

Arunima Singh, Ph.D.
Senior Editor
Nature Methods

Reviewer #1 (Remarks to the Author):

The authors have addressed all my previous comments with additional experiments, analyses, and explanations. Among the new experiments, I am particularly impressed by the demonstration of CHIMERYS' capabilities using the direct infusion dataset, which lacks precursor or fragment XICs. Despite not being initially designed for this type of data, CHIMERYS outperformed CsoDIAq. Congratulations on the excellent work and manuscript.

Fengchao

Reviewer #2 (Remarks to the Author):

The authors have adequately addressed comments and concerns.

Version 2:

Decision Letter:

6th Mar 2025

Dear Martin,

I am pleased to inform you that your Article, "Unifying the analysis of bottom-up proteomics data with CHIMERYS", has now been accepted for publication in Nature Methods. The received and accepted dates will be June 25, 2024 and March 6, 2025. This note is intended to let you know what to expect from us over the next month or so, and to let you know where to address any further questions.

Over the next few weeks, your paper will be copyedited to ensure that it conforms to Nature Methods style. Once your paper is typeset, you will receive an email with a link to choose the appropriate publishing options for your paper and our Author Services team will be in touch regarding any additional information that may be required. It is extremely important that you let us know now whether you will be difficult to contact over the next month. If this is the case, we ask that you send us the contact information (email, phone and fax) of someone who will be able to check the proofs and deal with any last-minute problems.

After the grant of rights is completed, you will receive a link to your electronic proof via email with a request to make any

corrections within 48 hours. If, when you receive your proof, you cannot meet this deadline, please inform us at rjsproduction@springernature.com immediately.

If you are active on Twitter/X, please e-mail me your and your coauthors' handles so that we may tag you when the paper is published.

Best regards,
Arunima

Arunima Singh, Ph.D.
Senior Editor
Nature Methods

** Visit the Springer Nature Editorial and Publishing website at http://editorial-jobs.springernature.com?utm_source=ejP_NMeth_email&utm_medium=ejP_NMeth_email&utm_campaign=ejp_Nmeth for more information about our career opportunities. If you have any questions please click [here](mailto:editorial.publishing.jobs@springernature.com).**

Open Access This Peer Review File is licensed under a Creative Commons Attribution 4.0 International License, which permits use, sharing, adaptation, distribution and reproduction in any medium or format, as long as you give appropriate credit to the original author(s) and the source, provide a link to the Creative Commons license, and indicate if changes were made. In cases where reviewers are anonymous, credit should be given to 'Anonymous Referee' and the source. The images or other third party material in this Peer Review File are included in the article's Creative Commons license, unless indicated otherwise in a credit line to the material. If material is not included in the article's Creative Commons license and your intended use is not permitted by statutory regulation or exceeds the permitted use, you will need to obtain permission directly from the copyright holder.

Reviewers' Comments:

The authors thank the reviewers for reading our manuscript and for the thoughtful suggestions and other comments made. In light of their advice, we have added new data, new data analyses, and new figure panels to the manuscript and changed text in many places to reflect these additions. This has made the manuscript stronger and we hope the reviewers now find it acceptable for publication.

Reviewer #1 (Remarks to the Author):

In this manuscript, the authors present a bottom-up proteomics data analysis software, CHIMERYYS. It applies a spectrum-centric approach to unify peptide identification and quantification for data-dependent acquisition (DDA), data-independent acquisition (DIA), and parallel reaction monitoring (PRM). In contrast to most traditional spectrum-centric approaches, CHIMERYYS heavily incorporates the deep-learning prediction of fragment intensities and retention time into the database searching and quantification procedures, which largely boosts the sensitivity, but also introduces limitations. To demonstrate the performance of CHIMERYYS, the authors used the existing DDA, DIA, and PRM data analysis tools as benchmarks. More importantly, the authors applied the entrapment database search to extensively evaluate the false discovery rate (FDR) of peptide identification performed by CHIMERYYS and other DIA tools. To benchmark the quantification results, they applied the LFQbench-type dataset to evaluate precision and accuracy. This manuscript has been written and organized. The proposed methods and CHIMERYYS software are promising. I am glad to see that people begin to care about quality over quantity in identification and quantification. However, several major and minor issues must be addressed.

The authors highly appreciate the comment on quality over quantity, which the authors believe is needed more than ever, and the notion that the proposed methods are promising.

Major issues

1. Unifying the analysis of DDA, DIA, and PRM is one of the major contributions of this work. From my point of view, this is not a significant scientific innovation. There have been tools, such as MSFragger and MSFragger-DIA, and MaxQuant and MaxLFQ, trying to unify the DDA and DIA data analysis. Furthermore, as far as I can see, there is no harm to analyzing DDA and DIA using different tools because people normally do not directly compare DDA with DIA in biological studies.

To clarify, the core scientific innovation described in this manuscript is the unifying algorithm implemented in CHIMERYYS. Unlike other software packages, CHIMERYYS analyzes spectra from DDA, DIA and PRM data, with or without LC separation, with the very same (!), spectrum-centric algorithm. The authors think this is of fundamental importance because the (tandem) MS spectrum is perhaps the most important part of a proteomic measurement and must, therefore, be the foundation for all data analysis. A central unique and innovative feature of CHIMERYYS is the capability of distributing the intensity of the same (shared) fragment ions that can arise from different peptides proportionally to these co-fragmented peptides via spectrum deconvolution. Other algorithms simply exclude these fragment ions, leading to loss of information and possibly error. As such, CHIMERYYS is a unifying algorithm that spans from the in-silico digest of the protein sequence (fasta) file and the generation of decoys via the scoring logic to FDR estimation.

Software packages such as MSFragger could be called unified in the sense that they bundle a collection of algorithms that each are specific for the analysis of certain types of data into a

single software package with a unified user interface (e.g. FragPipe). This is convenient for users but we note that e.g. analysing the same DIA data file with MSFragger or MSFragger-DIA leads to different results. This should, ideally, not be the case and we show that comparing the results generated by different search engines on the same data is further complicated by inconsistencies in the levels at which identifications are reported (PSMs, peptides or precursors) and at which FDR is controlled (e.g. peptide groups versus precursors).

Recent developments in proteomics highlight why a unifying algorithm is important. First, the latest generation of mass spectrometers (e.g. Orbitrap ASTRAL, timsTOF Ultra) are capable of generating MS2 spectra at a remarkable speed (>150 Hz). This, in essence, removes the difference between a DDA and DIA measurements. Second, this speed motivates scientists to use very short LC gradients (or no LC altogether) to support e.g. population-scale plasma proteome studies or single-cell investigations that both require very high sample throughput. DIA algorithms require extracted MS2 ion chromatograms (XICs) but, for very short LC gradients, many peptides are only detected in a single DIA scan, possibly with a very wide isolation window. This makes XIC-based interference removal error-prone. CHIMERYS overcomes these issues by applying the same, spectrum-centric algorithm to any kind of tandem MS data, which the authors think is a very substantial innovation.

We included a sentence in the Discussion clarifying the difference between unifying software packages and unifying algorithms and added a Supplementary Table describing the different reporting levels to facilitate future comparisons.

It would be better if the authors demonstrate some case studies that only the unified DDA and DIA analysis can do, while other approaches, such as using MSFragger for DDA and DIA-NN for DIA, will not be able to achieve.

In response to this suggestion, we reanalysed published data acquired by direct infusion shotgun proteomics analyses (DI-SPA) (Meyer et al., 2020). This is an interesting case because removing the LC separation step conceptually provides very high sample throughput and the plethora of (shared) fragment ions poses challenges to current data analysis concepts. We note that analyzing such data with MSFragger-DIA, while possible, violates the core assumption of detectable XICs (also true for MaxDIA and DIA-NN). Hence, bespoke, library-based software was written for this use case (MSPLIT-DIA by Wang et al., 2015 or CsoDIAq by Meyer et al., 2021). While MSPLIT-DIA removes PSMs the library entries of which are too similar to one another (projected cosine > 0.7), CsoDIAq – based on the concepts of MSPLIT-DIA – does not seem to have mechanisms that deal with shared fragment ions. We have added a new Figure 5 showing that CHIMERYS substantially outperforms CsoDIAq, the current state-of-the-art for DI-SPA data analysis. The up to three-fold increase in identifications unveils new biology in the form of previously undetected proteins and pathways, allowing scientists to dig deeper into DI-SPA data. To the best of our knowledge, CHIMERYS is the only ready-to-use, library-free algorithm with accurate FDR control and proper handling of shared fragment ions that can deal with any combination of arbitrary isolation window sizes and gradient lengths, from narrow/wide window direct infusion data via narrow/wide window DDA and narrow/wide window DIA data to narrow/wide window PRM data.

2. Although CHIMERYS uses the spectrum-centric approach, it has the limitations and disadvantages specific to the peptide-centric approach owing to its reliance on prediction. The limitations include limited modifications, peptide length ranges, and number of missed cleavages. These disadvantages are mostly related to the long runtime and additional requirements of graphics processing unit (GPU) hardware. The authors have discussed

those limitations in the discussion section: “A current limitation of CHIMERYYS in this context is that peptides carrying modifications that are not yet covered by the underlying deep learning model escape detection. It can be anticipated that this limitation will diminish over time as deep learning models start to emerge that are capable of generalizing to modifications or fragmentation methods that they have not yet been trained for.”. Unfortunately, there appears to be no ideal solution in the near future. Rescoring tools, such as MSBooster, overcome the modification limitation by stripping unsupported modifications during the prediction, and shifting the fragment peaks based on the modification masses after the prediction. Of course, this solution is not ideal because some modifications change the fragmentation and retention time. The authors should discuss more about the limitations and possible solutions if the desired deep learning models will not be available soon.

We note that CHIMERYYS already supports the most common PTMs, notably phosphorylation (S, T and Y), acetylation (K and protein N-terminus), ubiquitination (K) and oxidation (M). We have added Extended Data Figures 5G-H to highlight these capabilities and added a paragraph to the Supplementary Discussion on the currently limited set of supported PTMs. This discussion also covers that certain modifications substantially influence the fragmentation behavior of peptides and the resulting issues when trying to extrapolate to PTMs not used for model training or when shifting fragment peaks based on modification masses. One instructive example are citrullinated (Cit) peptides that a Cit-unaware model falsely places closer to decoys than to targets. In addition, we also discuss alternative embeddings found in the literature that would potentially allow extrapolation to “unseen” modifications in the future.

3. The authors did not compare runtimes among the CHIMERYYS and other tools. This is an important benchmark because 1) the data scale is becoming increasingly larger with the development of mass spectrometry and sample preparation techniques, and 2) CHIMERYYS is computationally intensive because it performs deep-learning predictions. The authors should perform the runtime comparison using the tools mentioned in Figure 1 and 3.

We have added new Extended Data Figures 3D and 9A, where we compare the runtime of CHIMERYYS to other tools. Briefly, while CHIMERYYS is not the fastest of the search engines we compared, it is well capable of analyzing data in less time than it took to acquire it. As such, it is well suited for analyzing the plethora of proteomic data generated today.

4. The current version of CHIMERYYS only supports mass spectrometry data generated by the Thermo Fisher mass spectrometer. Do the authors have any plans to support the data from Bruker (especially timsTOF series), SCIEX, and Waters mass spectrometers?

Yes, we are in the process of implementing support for mzML and other vendor-specific formats, which will enable the analysis of proteomic data acquired on any mass spectrometer, including Bruker instruments. We added a sentence to the discussion to reflect this.

5. In the Introduction section, the authors reviewed the peptide-centric approach for DIA data analysis, but missed the spectrum-centric approaches include DIA-Umpire (<https://doi.org/10.1038/nmeth.3255>), PECAN (<https://doi.org/10.1038/nmeth.4390>), MSFragger-DIA (<https://doi.org/10.1038/s41467-023-39869-5>), and diaTracer (<https://doi.org/10.1101/2024.05.25.595875>). It would be better to review them because the spectrum-centric approach is also what CHIMERYYS is using.

We apologize for this omission. We have included a section on library-free DIA data analysis using software such as DIA-Umpire, PECAN, MSFragger-DIA and diaTracer, as well as the corresponding references in the introduction.

6. The authors need to describe how they calculated the “CHIMERYYS coefficient” because I could not find a clear description in either the Results or Online Methods section. In the Results section, they wrote “Apart from peptide identification, it also derives spectrum-centric quantitative information in the form of CHIMERYYS coefficients, which can be interpreted as the interference-corrected total ion current for a given peptide in this MS2 spectrum (Online Methods). If none of the matched fragments for a peptide are shared with another peptide and the predicted MS2 spectrum matches perfectly to the experimental one, the coefficient is the sum of all matched fragment ions in the experimental MS2 spectrum. Hence, tracing the coefficient along retention time generates a pseudo-extracted-ion-chromatogram (XICs) that can be used to perform (relative) quantification of peptides based on their MS2 signal in PRM and DIA data.”. What about there are matched fragments shared with another peptides? In addition, how can XIC be generated for DDA data with no fragment XICs?

We apologize if our initial description was not clear. Regarding the question on matched fragments shared between peptides: essentially, CHIMERYYS treats chimeric spectra as linear combinations of pure spectra. Here, predicted spectra for high-scoring candidate peptides serve as the source of pure spectra. One can represent the predicted spectra of all candidate peptides as a $P \times M$ matrix of intensities $I_{p,m}$, where $p = 1, 2, \dots, P$ is an index over the predicted spectra and $m = 1, 2, \dots, M$ is an index over mass channels. The mass channels are defined by the peaks of the experimental spectrum that were within the recalibrated fragment ion match tolerance of at least one fragment ion from the collection of predicted spectra, as well as all unmatched fragment ions from said collection. As such, if two peaks from two different predicted spectra match to the same experimental peak, they will have the same mass channel. This is how CHIMERYYS handles shared fragment ions. The CHIMERYYS coefficients β_p then define a combined spectrum based on the above-mentioned matrix as the linear combination of the predicted spectra, scaled by the corresponding coefficients. The coefficients are chosen such that they minimize the LASSO objective function, which strikes a balance between the number of peptides with non-zero coefficients and the mean squared error between the combined spectrum and the experimental spectrum. We included additional information on the calculation of the CHIMERYYS coefficients in the Online Methods section.

In the section of the main text you cited above, we mention that “tracing the coefficient along retention time generates a pseudo-extracted-ion-chromatogram (XICs) that can be used to perform (relative) quantification of peptides based on their MS2 signal in PRM and DIA data.” We added the information that this is generally not possible for DDA data.

7. In the Results section, the authors wrote “Briefly, predicted MS2 spectra from precursors with predicted retention times that fall within a data-dependent retention time window and precursor isotope envelopes that (partially) overlap with the isolation window are compared to experimental MS2 spectra.”. Please elaborate more about the “data-dependent retention time window”.

In the Online Methods, we write “In the main search, the above-described scoring functions are executed using the optimized settings and prediction parameters, albeit now also filtering candidate peptides based on their predicted retention time. CHIMERYYS uses retention time tolerance windows that would allow the identification of all peptides confidently identified in the initial search.” We added two sentences after this one clarifying the details of this calculation. Briefly, we calculate the absolute difference of experimental and predicted retention times after refinement learning for all PSMs identified at 1% FDR in the initial search. The 100% quantile of these differences is the basis for the initial retention time

window (+/- 100% quantile of absolute differences between experimental and predicted RT), which is then expanded further by multiplying it with a security factor of 2.5.

8. In Figure 1C, have the authors checked whether these peptides' predicted retention time is within a reasonable tolerance of the scan's observed retention time?

On average, the predicted retention time differs by 1.14 minutes from the observed retention time, which corresponds to less than 1% deviation relative to the gradient length of 120 minutes. We added the predicted retention times for these peptides to the Results section.

9. In Figure 3D, what are the criteria for selecting the precursors for the top and bottom panels? It might be unfair if pick a high-scored one identified by both Spectronaut and CHIMERYs while pick a low-scored one identified only by Spectronaut. Perhaps the authors should pick the lowest-scored one identified by both Spectronaut and CHIMERYs for the top panel and the highest-scored one identified only by Spectronaut for the bottom panel. It will support the authors' argument if the top one still has a higher quality than the lower one.

Thank you for raising this point. To address this comment and to clarify the selection of the precursors in Figure 3D, we rephrased the Results:

"[...] CHIMERYs is more rigorous in the inclusion of fragment ions with very low intensity. The latter is illustrated in Figure 3D. The top panel shows fragment ion chromatograms exported from Spectronaut for a precursor confidently identified by all three search engines. The bottom panel shows fragment ion chromatograms exported from Spectronaut for the highest-scoring, Spectronaut unique precursor that was entirely based on fragment ions with an $F.P_{eakArea} \leq 1$. Inspection of the corresponding raw data revealed that these fragment ions are missing in the relevant retention time range (Extended Data Figure 9E, see also Supplementary Information). Both precursors were identified by Spectronaut with comparable scores and posterior error probabilities."

Further, we rephrased the corresponding figure legend:

"[...] (D) Example fragment ion XICs exported from Spectronaut for two high-scoring precursors with similar confidence (all six library fragments are shown). The top one is also identified by CHIMERYs, while the bottom one is not. Spectronaut identifies some precursors based entirely on fragment ions with intensities below or equal to 1 according to their report. The bottom panel shows the highest-scoring one of these precursors."

Hence, the criteria for selecting the precursors are as follows: the bottom precursor is the highest-scoring precursor from Spectronaut where all corresponding fragment ions have an $F.P_{eakArea} \leq 1$. The top precursor is one with a similar score where all corresponding fragment ions have an $F.P_{eakArea} > 1$ that was also identified by CHIMERYs.

10. In the Results section, the authors wrote "To avoid differences in quantification due to different methods for determining peak integration borders, we compared the three algorithms based on their implementation of peak apex quantification.". I do not quite understand how they "compered the three algorithms based on their implementation of peak apex quantification". Did the authors re-implement the peak quantification algorithms of DIA-NN and Spectronaut? However, to the best of my knowledge, these algorithms are not publicly available.

We apologize that this point was not phrased more clearly. We did not re-implement the peak quantification algorithms of DIA-NN and Spectronaut but instead simply used their corresponding settings (Quantification strategy = Peak height for DIA-NN and Quantity Type = Height for Spectronaut) as described in the Online Methods. For CHIMERYs, the

calculation of the peak apex intensity is also described in the Online Methods: “[...] quantification of PRM and DIA data is performed by [...] using the highest CHIMERY5 coefficient within the integration borders as the elution peak apex intensity.”. We modified the confusing sentence in the Results section to guide the reader to the Online Methods.

11. In the Online Method section “data preprocessing-retention time prediction”, the authors should clarify whether CHIMERY5 1) performs retention time alignment; and 2) uses apex retention time or the top-ranked PSM’s precursor retention time.

The section entitled “Data preprocessing – retention time prediction” describes how training data is preprocessed for the training of our INFERY5 retention time base model, which is independent of the refinement learning that happens in CHIMERY5. We added a sentence to this section highlighting that it refers to the training of our INFERY5 base models and not the processing within CHIMERY5.

1) Within CHIMERY5, we do not perform classic retention time alignment. Instead, we apply refinement learning to each raw file using INFERY5’ base retention time model as a starting point and the results from the first search. This results in distinct retention time models for each raw file that minimize the difference between predicted and experimental retention times. Comparing retention times between raw files then involves predictions with these raw file-specific models.

2) During refinement learning, for each raw file, we filter the results of the coarse first search to 1% FDR, followed by the removal of decoys, PSMs with a normalized spectral contrast angle smaller than 0.7 and PSMs for which the number of matched peaks divided by the number of predicted peaks is smaller than 0.6. We then select the top-ranked PSM per spectrum by LDA score and then select the top 10,000 PSMs per raw file by LDA score as the foundation for our retention time refinement learning. As such, we use the retention times of at least one (often multiple) top-ranked PSM per precursor for refinement learning. We expanded the corresponding section of the Online Methods accordingly.

Minor issues

1. The resolution of the figures seems to be lower than expected. It would be better if the authors double check them.

We apologize for the inconvenience. For this revision, we made sure to preserve all figures at high resolution.

2. In the Online Methods section, the link of Thermo Fisher’s RawFileReader (<http://planetorbitrap.com/rawfilereader>) does not seem correct. As far as I know, it should be <https://github.com/thermofisher/lsms/RawFileReader>.

Thank you for pointing this out. We have corrected it in the manuscript.

3. In the Online Methods section, the authors wrote “Searches were performed using MSFragger v20.0 using the "Default" workflow. Additionally, MSFragger v21.1 was utilized with the "WWA" workflow and "DDA+" data type...”. “v20.0” and “v21.1” are FragPipe’s version numbers. The authors should also list the versions of MSFragger and Philosopher since they are used alone with FragPipe in the workflow.

We now included the version numbers of MSFragger and Philosopher in the Online Methods.

4. In the “Code availability” section, the authors wrote “The custom R scripts used for data analysis are available upon request.”. To increase the transparency of this scientific manuscript, it would be better to upload the scripts to GitHub.

We started to upload the custom R scripts used for data analysis to GitHub and amended the Code availability section accordingly. All scripts will be uploaded before publication.

Reviewer #2 (Remarks to the Author):

The authors of the manuscript have successfully shown the utility of the CHIMERY5 algorithm for the identification and quantification of peptides from numerous bottom-up proteomics data types. The spectrum-centric approach, which integrates deep learning predictions with a parsimonious identification algorithm, represents a novel and promising advancement in the field. The data and methodology presented are comprehensive and robust, though there are a few areas that warrant revision.

We thank the reviewer for their positive comments. We hope that our responses below and our changes to the manuscript satisfy the need for revision.

Supplemental figure 4A-C: This figure is used to interpret results unique to CHIMERY5. However, the claim that CHIMERY5 produces high-quality identifications while CHIMERY5 specific results have a lower average spectral contrast angle requires further explanation. The authors should clarify why this metric is used as a supporting argument for their conclusion.

We added a paragraph on this topic to the Supplementary Discussion. Briefly, CHIMERY5 unique identifications have lower MS intensity than those shared with other search engines (Extended Data Figure 4A). As such, they have fewer matched fragment ions than shared peptides (median of 10 versus 17, Extended Data Figure 4B) but still high normalized spectral contrast angles (median of 0.69 vs 0.85, Extended Data Figure 4C, see also Supplementary Discussion). Hence, they are readily distinguished from decoys using mokapot’s SVM Score that aggregates the CHIMERY5 score set (Extended Data Figure 4D). We modified Extended Data Figure 4 to highlight this point in more detail and changed the corresponding sentence in the Results section. For further details, we refer the reviewer to the Supplementary Discussion.

We also note that the median SA for CHIMERY5 unique identifications of 0.69 is much higher than the median SA for decoy identifications of 0.44. As shown in Extended Data Figure 2C of the original ProSIT publication (Gessulat et al., 2019), an SA of 0.7 corresponds to a Pearson correlation of 0.88, which is the average prediction accuracy of MS2PIP (Declercq et al., 2022). Therefore, CHIMERY5 unique identifications are of very high quality.

Optimization of Data Acquisition: The conclusions in the second paragraph of the section “Optimizing data acquisition with deconvolution in mind” would benefit from clarification. The discussion on gradually widening isolation windows, the resulting loss in peptide/protein sensitivity in CHIMERY5 results, and the subsequent claim that CHIMERY5 can counteract this effect, is somewhat unclear. It is not immediately evident which effect is being mitigated, especially since the described loss in sensitivity is a direct outcome of a CHIMERY5 search.

We apologize for the confusion and hope to convey this point more clearly now. In the final sentence of this paragraph (“CHIMERY5 can counteract this effect [...]”), “this effect” refers

to the sentence, where we discuss that “[...] in single-cell proteomics [...] extended injection times enhance sensitivity but result in fewer MS2 scans”. We wanted to highlight that in SCP, where extended injection times result in fewer MS2 scans with a higher signal-to-noise ratio, the accompanying loss in peptide identifications with traditional search engines can be counteracted by CHIMERY5. This is because unlike CHIMERY5, traditional search engines only have very limited capacity of identifying multiple peptides in a single MS2 spectrum. This statement is disconnected from the first half of the paragraph, where we refer to experiments where the isolation window was varied between 1.4 Th and 20.4 Th for measurements of the same, high-load sample. There, we observed a loss in peptide/protein sensitivity in both CHIMERY5, MSFragger and Sequest HT as the isolation window was increased and we give an explanation for this observation. Briefly, in high-load samples like these, this is likely due to the AGC limit, which – together with the dynamic range of MS2 spectra – limits the number of peptides in chimeric spectra with a sufficient number of detectable fragment ions. The number of unique peptide (and protein) identifications reached its maximum already at a window size of 3.4 Th for this specific dataset and substantially decreased for larger isolation windows, likely due to the fact that more and more PSMs were from the same, high-abundant peptides that were now co-isolated more often. We have revised this section of the manuscript to improve clarity.

Comparison with Other Software: Several questions arise in the comparison of CHIMERY5 to other software. First, the rationale for selecting Sequest HT as the standard for comparison after an initial comparison with multiple software tools should be better explained. Additionally, some of the software versions used in the comparison appear to be several years out of date—the most up to date version of these software should be used to ensure a fair comparison.

The rationale for selecting Sequest HT was that protein grouping, as well as peptide- and protein-level FDR estimation for Sequest HT and CHIMERY5 is performed by Proteome Discoverer, improving the comparability of results at the peptide and protein level. However, we have now also added a comparison to MSFragger, the second best performing software after CHIMERY5, to Extended Data Figure 5. We have added an explanation of our choice to the Results section. In addition, we updated the results of Figure 1E with results from the most recent versions of each of the search engines.

CHIMERY5 Coefficient: The CHIMERY5 coefficient is described under ideal conditions with no shared fragment ions. The authors should address how this calculation is modified in the presence of shared fragment ions and how such conditions might impact quantification.

We included additional information on the calculation of the CHIMERY5 coefficients and how shared fragment ions impact quantification in the Online Methods section. In essence, the calculation of the CHIMERY5 coefficient is identical, no matter how many fragment ions are shared between co-eluting peptides. In the Results section “Accurate peptide quantification from chimeric PRM and DIA spectra”, we only describe the ideal scenario to give the reader an intuition for its interpretation.

Server Setup for Predictions: The decision to set up a server for predictions, rather than allowing them to be performed locally, is not sufficiently justified. Providing a rationale for this choice would enhance the clarity of the methodology.

We decided to set up a cluster of servers for predictions, rather than allowing them to be performed locally in order to reduce the runtime of CHIMERY5 and facilitate supporting the software in the long-term. CHIMERY5 needs to predict the MS2 spectrum for a given peptide up to twice per raw file: once for the first search and once – after collision energy

(CE) recalibration – for the main search. This is because our deep learning model uses CE as an additional input, since it substantially influences fragmentation and can vary between mass spectrometers and even over time for the same mass spectrometer (Gessulat et al., 2019). However, in particular for large search spaces commonly encountered when working with dynamic modifications such as phosphorylation, predicting the entire spectral library up to twice per raw file is computationally expensive (2x >90M precursors per raw file as we need to predict multiple charge states). With ~2,000 predictions per second on a CPU and ~30,000 predictions per second on a GPU, this means that scientists would have to wait either 25h or 1.6h per raw file for these predictions to be performed using a single CPU or GPU, respectively. That is why we decided for GPU-based predictions in the cloud, where we can additionally scale out to multiple GPU-containing servers. This also allowed us to standardize and streamline the setup of these servers, which is quite important and often difficult, since successful predictions depend for example on matching GPU hardware and CUDA versions. In the future, we would like to also be able to perform predictions locally. However, for this to be feasible, current machine learning models need to become much faster on CPUs. INFERYS Rescoring (Zolg et al., 2021) can run locally, because it only predicts the top 10 peptides per spectrum from Sequest HT, which drastically reduces the total number of predictions required. We have added a sentence describing our reasoning to the “Setup” paragraph in the Online Methods and a more extensive explanation to the Supplementary Discussion.

Clarity and Context:

- The use of the phrase “en passant” is unconventional and may be obscure to many readers. Given its primary definition relates to a chess move, the sentence in which it is used may be confusing. Consider replacing it with an English term or a more commonly understood phrase.

In order to avoid confusion, we have rephrased the sentence in question.

- The introduction discusses the subtractive approach to chimeric identifications and its limitations but does not address the multiplicative approach, where fragment ions are not removed for an additional search with a different precursor. This omission is further compounded in the results section, where the current handling of chimeric spectra is described as error-prone due to fragment ion peaks shared by multiple peptides.

We rephrased the sentence in the introduction describing the multiplicative approach and its downsides to contrast the two methods. We also refer to them in the respective sentence of the Results section.

- The statement, “A central challenge for data analysis lies in the fact that most MS2 spectra are chimeric, i.e., they contain more than one peptide because LC-MS/MS systems cannot fully separate the vast number of peptides resulting from whole proteome enzymatic digestion,” is foundational to the paper but lacks citation and empirical support. The claim is supported only by Supplemental Figure 1, which is based on search results rather than the spectra themselves.

We added multiple citations to the introduction reflecting that narrow-window DDA spectra are often chimeric and rephrased the corresponding sentence accordingly.

- The conclusions in the first paragraph of the “Comparison to other DIA search engines” section could be revised for greater clarity. The process by which the “Reduction to a comparable level” was achieved and the definition of “data completeness” are not fully explained.

We revised the first paragraph of the “Comparison to other DIA search engines” section for greater clarity and explain the definition of “data completeness” and the process by which the “Reduction to a comparable level” was achieved. Briefly, Extended Data Figure 8 shows that neither DIA-NN, nor Spectronaut properly control FDR in the run-specific context by comparing eFDR to the algorithm-dependent, self-reported FDR. The “Reduction to a comparable level” was achieved by filtering at 1% eFDR in addition to 1% algorithm-dependent, self-reported FDR, thereby ensuring proper FDR control in the run-specific context. The number of precursors identified in two (Figure 3A) or three (Extended Data Figure 9B and F) out of three replicates relative to the overall number of identifications was used as a measure for data completeness (i.e. the height of the gray bar relative to the height of the orange bar).

Reviewer #2 (Remarks on code availability):

Percolator Mimic was readily available and easily understandable. Their version of mokapot requires a fair amount of github and coding knowledge to access.

We have merged our changes to mokapot back to the main branch. Once released by the mokapot maintainers, they will be as easily accessible as prior versions of mokapot.

Reviewers' Comments:

The authors thank the reviewers for reading our manuscript and rebuttal letter. We are happy to hear that we adequately addressed all their comments and concerns.

Reviewer #1:

Remarks to the Author:

The authors have addressed all my previous comments with additional experiments, analyses, and explanations. Among the new experiments, I am particularly impressed by the demonstration of CHIMERYs' capabilities using the direct infusion dataset, which lacks precursor or fragment XICs. Despite not being initially designed for this type of data, CHIMERYs outperformed CsoDIAq. Congratulations on the excellent work and manuscript.

Reviewer #2:

Remarks to the Author:

The authors have adequately addressed comments and concerns.